# Application of the *ps*−Version of the Finite Element Method to the Analysis of Laminated Shells

**DOI:** 10.3390/ma16041395

**Published:** 2023-02-07

**Authors:** Cheng Angelo Yan, Riccardo Vescovini

**Affiliations:** Dipartimento di Scienze e Tecnologie Aerospaziali, Politecnico di Milano, Via La Masa 34, 20156 Milano, Italy

**Keywords:** finite element method, numerical methods, thin shells, variable-stiffness structures

## Abstract

The development of accurate and efficient numerical methods is of crucial importance for the analysis and design of composite structures. This is even more true in the presence of variable stiffness (VS) configurations, where intricate load paths can be responsible for complex and localized stress profiles. In this work, we present the ps−version of the finite elements method (ps−FEM), a novel FE approach which can perform global/local analysis through different refinement strategies efficiently and easily. Within this framework, the global behavior is captured through a p−refinement by increasing the polynomial order of the elements. For the local one, a mesh−superposition technique, called s−refinement, is used to improve locally the solution by defining a local/fine mesh overlaid to the global/coarse one. The combination of p− and s−refinements enables us to achieve excellent accuracy−to−cost ratios. This paper aims to present the numerical formulation and the implementation aspects of this novel approach to VS composite shell analysis. Numerical tests are reported to illustrate the potential of the method. The results provide a clear insight of its potential to guarantee fast convergence and easy mesh refinement where needed.

## 1. Introduction

The anisotropic nature of composite laminates is responsible for structural responses characterized by complex couplings between extension, bending, torsion, and shearing modes [1]. Such effects are particularly relevant when laminates with curvilinear reinforcements [2,3,4,5] are of concern.

These effects require appropriate numerical tools to be available to conduct accurate and reliable analyses. This is even more true as composite thin shells are increasingly used in primary load−carrying structural components. Hence, the interest of the designers is not restricted to the global response, but includes also local effects, such as stress profiles in critical regions due to geometric discontinuities, material interphases, and areas affected by load introduction. In this framework, the finite element method (FEM) is the most commonly used tool for the analysis and design of composite structures. The main advantages can be found in the high flexibility of modeling complex domains, the possibility of considering arbitrary material properties, and handling different loading/boundary conditions.

Over the years, different strategies have been developed to maximize the accuracy of the FE solution, while keeping the computational cost at a minimum. The classical approaches to improve accuracy are represented by the h− and p−refinements [6]. In the former, accuracy is increased by reducing the mesh size *h*, while the interpolation order of the elements *p* is fixed [7]. In the latter, the polynomial degree *p* is increased, while the mesh resolution *h* is left unchanged [8]. The h− and p−refinements provide two systematic ways to improve the precision of the FE solution and yield excellent results when global quantities are of interest. Different applications of these strategies to advanced composite shells and plates can be found in the literature, such as sandwich plates [9], functionally graded laminates [10,11], shell structures with orthogonal periodic configurations [12], cylindrical shells made of metamaterials with spatially variable elastic properties [13], and variable stiffness composites [14,15,16,17].

Local refinement strategies are needed whenever focus is required on the response in specific regions. Adaptive h−refinement procedures have been proposed in the literature, e.g., [18,19,20]. In these works, a posteriori error estimators are coupled with the h−refinement strategy to generate adaptable meshes locally refined in the most critical zones. The hp−refinement approach [21] represents an extension of the adaptive h−refinement where an increase of the interpolation order of elements (p−refinement) is added on top of a suitable mesh adaptation (h−refinement). A simultaneous increase of the order *p* combined with a graded increase of the mesh resolution *h* yields exponential convergence rate for non−smooth problems, where steep gradients or singular points are present [22,23].

One of the challenges associated with simultaneous h− and p−refinements relates to the complexity in generating an h−adaptable mesh. Typical procedures to cope with this problem involve the adoption of transition elements [24,25,26] or sophisticated multi-constraints approaches [27].

An alternative and simpler way to perform local mesh refinements is given by mesh superposition techniques. One of the first works in this field was by Mote [28], who employed the Ritz method to capture the global deflections of the structure, while a superposed local FE model was proposed for capturing stress concentrations.

The spectral overlay finite element method (SOFEM) proposed by Belytschko [29] is another example of an FE scheme based on solution superposition, where local refinements are performed by a spectral overlay on an FE mesh. A generalization of SOFEM is represented by the superposition−version, or s−version, of the finite element method (s−FEM) proposed by Fish [30]. In the s−FEM the solution accuracy is locally enhanced by superposing one or more high−order local meshes onto a global one. This extension of the FEM has been applied successfully for linear static, frequency, and buckling analysis of laminated composite plates and shells [31,32,33,34], stress analysis of laminated smart structures [35], microscopic stress analysis of heterogeneous materials [36], and dynamic crack analysis in steel structures [37].

A further superposition refinement approach is implemented in the hp−d−version of the finite element method (hp−d−FEM) developed by Rank [38]. The hp−d−FEM improves the solution efficiency by representing the smooth part of the solution with a coarse/high−order global mesh, while the non−smooth features are resolved by one or more [39] fine/low−order local meshes. The method is applied for solving linear and nonlinear elasticity problems with singularities in [39,40]. The multi−level hp−method introduced by Zander et al. [41] recovers the idea of the hp−d−FEM where the superposition meshes used for the local refinement employ high−order instead of low−order elements. This approach permits to achieve much faster convergence rates, as shown in [42,43], where the method is used for solving fracture mechanics, linear elastodynamics, and shell problems.

All the previous works have shown the potential of mesh superposition techniques to achieve refined solutions while keeping at minimum the implementation efforts. The present work aims at recovering such techniques to realize an efficient global/local formulation of the FEM, here denoted as the ps−version of the finite element method (ps−FEM), for the analysis of laminated composite shells. More specifically, the FE framework presented herein exploits a p−refinement strategy, based on the hierarchic polynomial space first formulated by [8], to capture the global response of the laminate. For the resolution of the local solution features, the p−refinement is combined with the mesh superposition refinement approach, or s−refinement, developed by [30,41].

Differently from existing works in the literature, superposition techniques are employed here for the first time for the analysis of composite shells, including straight−fiber and VS configurations.

The outline of the paper is as follows: the shell mathematical model is presented in Section 2; Section 3 discusses the formulation and implementation aspects of the proposed FEM; in Section 4, the method is applied to several test cases involving laminated shells. Finally, the main findings of this study are summarized in Section 5.

## 2. Theoretical Model

This section illustrates the theoretical framework for the analysis of laminated composite shells. In the first part, the shell geometry is discussed by presenting the relevant equations employed next in the finite element approximation. Then, the constitutive law for handling the case of curvilinear fiber paths is presented. Finally, the shell governing equations are derived in the context of a total Lagrangian description.

### 2.1. Geometry Description

The geometric description of shells is developed under the theory of surfaces, which provides a general framework to define panels with arbitrary configurations and curvatures.

The reference surface Ω is assumed to be coinciding with the shell’s midsurface and is parametrized by two arc−length coordinates $\xi_{1}\in \left[0,a\right]$ and $\xi_{2}\in \left[0,b\right]$, where *a* and *b* are the curvilinear lengths of the shell edges. The normal direction to Ω is parametrized by the coordinate ζ∈[−t/2,+t/2], where *t* is the thickness of the shell. It is assumed that the triad $\left(\xi_{1},\xi_{2},\zeta \right)$ forms an orthogonal curvilinear coordinate system such that $\xi_{1}$ and $\xi_{2}$ are aligned with the principal lines of curvature on the midsurface, as shown in Figure 1.

Considering a global reference system (O,x,y,z), any point on Ω can be described by the position vector r:(1)$$r\left(\xi_{1},\xi_{2}\right)=r_{x}\left(\xi_{1},\xi_{2}\right)e_{x}+r_{y}\left(\xi_{1},\xi_{2}\right)e_{y}+r_{z}\left(\xi_{1},\xi_{2}\right)e_{z}$$
where $r_{x}$, $r_{y}$ and $r_{z}$ are definite, continuous, single-valued functions of $\xi_{1}$ and $\xi_{2}$ representing the components of r on the global reference system with unit directors $e_{x}$, $e_{y}$ and $e_{z}$.

The tangent vectors on Ω to the two coordinate lines, $\xi_{1}$ and $\xi_{2}$, are obtained by differentiation of Equation (Equation 1):(2)$$g_{\alpha }=\frac{\partial r}{\partial \xi_{\alpha }}=r_{,\alpha }\;\mathrm{for}\;\alpha =1,2$$
where the comma denotes differentiation with respect to the coordinate $\xi_{\alpha }$.

Accordingly, the unit normal to the reference surface is defined as:(3)$$n=\frac{g_{1}\times g_{2}}{\left|g_{1}\times g_{2}\right|}$$
where × denotes the cross product and |·| the Euclidean norm. The unit normal in Equation (Equation 3) is assumed to be common to all the shell laminas $\Omega^{\left(\zeta \right)}$, see Figure 1.

Considering a lamina at a distance ζ from Ω, the position vector of the generic point on $\Omega^{\left(\zeta \right)}$ is expressed as:(4)$$r^{\left(\zeta \right)}\left(\xi_{1},\xi_{2},\zeta \right)=r\left(\xi_{1},\xi_{2}\right)+\zeta n\left(\xi_{1},\xi_{2}\right)$$
while the corresponding tangent vectors along the curvilinear lines $\xi_{1}$ and $\xi_{2}$ are:(5)$$g_{\alpha }^{\left(\zeta \right)}=r_{,\alpha }^{\left(\zeta \right)}=r_{,\alpha }+\zeta n_{,\alpha }\;\mathrm{for}\;\alpha =1,2$$

By application of the Weingarten–Gauss relations [1]:(6)$$n_{,\alpha }=\frac{g_{\alpha }}{R_{\alpha }}$$
the following relation between the tangent vectors on $\Omega^{\left(\zeta \right)}$ and Ω holds:(7)$$g_{\alpha }^{\left(\zeta \right)}=\left(1+\frac{\zeta }{R_{\alpha }}\right)g_{\alpha }\;\mathrm{for}\;\alpha =1,2$$
where $R_{\alpha }$ are the principal radii of curvature of the reference surface and are defined as [44]:(8)$$R_{\alpha }=-\frac{g_{\alpha }\cdot g_{\alpha }}{g_{\alpha ,\alpha }\cdot n}\;\mathrm{for}\;\alpha =1,2$$
where · denotes the dot product. In general, the two principal radii of curvatures $R_{1}$ and $R_{2}$ are not constant and can be a function of the coordinate lines $\xi_{1}$ and $\xi_{2}$.

Referring to Equation (Equation 4), the differential position vector is given as:(9)$$dr^{\left(\zeta \right)}=r_{,1}^{\left(\zeta \right)}d\xi_{1}+r_{,2}^{\left(\zeta \right)}d\xi_{2}+r_{,\zeta }^{\left(\zeta \right)}d\zeta =g_{1}^{\left(\zeta \right)}d\xi_{1}+g_{2}^{\left(\zeta \right)}d\xi_{2}+nd\zeta$$

From Equation (Equation 9), the square distance dl between two arbitrary points in the shell volume $P(\xi_{1},\xi_{2},\zeta )$ and $P^{\prime }\left(\xi_{1}+d\xi_{1},\xi_{2}+d\xi_{2},\zeta +d\zeta \right)$ is computed as:(10)$${\left(dl\right)}^{2}=dr^{\left(\zeta \right)}\cdot dr^{\left(\zeta \right)}={\left(a_{1}^{\left(\zeta \right)}d\xi_{1}\right)}^{2}+{\left(a_{2}^{\left(\zeta \right)}d\xi_{2}\right)}^{2}+{\left(d\zeta \right)}^{2}$$
where $a_{\alpha }^{\left(\zeta \right)}$ are the Lamè parameters defined as:(11)$$a_{\alpha }^{\left(\zeta \right)}=\sqrt{g_{\alpha }^{\left(\zeta \right)}\cdot g_{\alpha }^{\left(\zeta \right)}}=\left(1+\frac{\zeta }{R_{\alpha }}\right)a_{\alpha }\;\mathrm{for}\;\alpha =1,2$$
with:(12)$$a_{\alpha }=\sqrt{g_{\alpha }\cdot g_{\alpha }}\;\mathrm{for}\;\alpha =1,2$$

For the development of the shell mathematical model, the geometric measures are expressed in terms of reference surface Ω quantities. Accordingly, the element lines and cross section areas along the coordinates $\xi_{1}$ and $\xi_{2}$ are given as:(13)$$\begin{array}{cc} \; & dl_{\alpha }=a_{\alpha }^{\left(\zeta \right)}d\xi_{\alpha }=\left(1+\frac{\zeta }{R_{\alpha }}\right)a_{\alpha }d\xi_{\alpha }\\ \; & dA_{\alpha }=dl_{\alpha }d\zeta =\left(1+\frac{\zeta }{R_{\alpha }}\right)a_{\alpha }d\xi_{\alpha }d\zeta \;\mathrm{for}\;\alpha =1,2 \end{array}$$
while the element surface and volume are given by:(14)$$\begin{array}{cc} \; & d\Omega =\left|r_{,1}^{\left(\zeta \right)}d\xi_{1}\times r_{,2}^{\left(\zeta \right)}d\xi_{2}\right|=a_{1}a_{2}\left(1+\frac{\zeta }{R_{1}}\right)\left(1+\frac{\zeta }{R_{2}}\right)d\xi_{1}d\xi_{2}\\ \; & dV=\left(r_{,1}^{\left(\zeta \right)}d\xi_{1}\times r_{,2}^{\left(\zeta \right)}d\xi_{2}\right)\cdot r_{,\zeta }^{\left(\zeta \right)}d\zeta =a_{1}a_{2}\left(1+\frac{\zeta }{R_{1}}\right)\left(1+\frac{\zeta }{R_{2}}\right)d\xi_{1}d\xi_{2}d\zeta \end{array}$$

The geometric formulation presented here furnishes the background to develop the finite elements discussed next, where plates and shells with variable curvature can be taken into account.

### 2.2. Curvilinear Fiber Path Description

The shell elements considered here can be applied to the analysis of isotropic and composite structures. For generality purposes, the model is developed to allow non−uniform elastic properties to be considered along the surface. Hence, new configurations such as variable stiffness (VS) laminates can be studied within the proposed framework. Specifically, a laminate with VS properties is achieved by stacking together plies with curvilinear reinforcing fibers [2]. Different strategies have been proposed in the literature for describing the path of the fibers. Without loss of generality, the approach considered here relies on the Lagrange polynomials introduced in [4]. According to this mathematical formulation, the curvilinear paths are described by a set of parameters $T_{mn}$ according to:(15)$$\theta \left(\xi_{1},\xi_{2}\right)=\sum_{m=0}^{M-1}\sum_{n=0}^{N-1}T_{mn}\prod_{n\neq i}\frac{\left(\xi_{1}-\xi_{1}^{\left(i\right)}\right)}{\left(\xi_{1}^{\left(n\right)}-\xi_{1}^{\left(i\right)}\right)}\cdot \prod_{m\neq j}\frac{\left(\xi_{2}-\xi_{2}^{\left(j\right)}\right)}{\left(\xi_{2}^{\left(m\right)}-\xi_{2}^{\left(j\right)}\right)}$$
where $\left(\xi_{1}^{\left(n\right)},\xi_{2}^{\left(m\right)}\right)$ are pre−selected control points over a uniform grid N×M in $\left[0,a\right]\times \left[0,b\right]$. From Equation (Equation 15), it is possible to understand the parameters $T_{mn}$ as the fiber orientation angle at the control points, as one can see that $\theta \left(\xi_{1}^{\left(n\right)},\xi_{2}^{\left(m\right)}\right)=T_{mn}$. The information of fiber orientation on a the generic VS ply can thus be defined through a matrix $T\in R^{M\times N}$. This enables us to specify VS lamination sequences very concisely as $\left[T_{1}/T_{2}/\cdots /T_{N_{p}}\right]$, where $N_{p}$ stands for the number of plies.

An alternative formulation is also possible where the control points are defined at a quarter of the laminate and the resulting fiber orientation distribution is reflected to the rest of the domain. In this case, the expression of Equation (Equation 15) modifies as:(16)$$\theta \left(\xi_{1},\xi_{2}\right)=\sum_{m=0}^{M-1}\sum_{n=0}^{N-1}T_{mn}\prod_{n\neq i}\frac{\left(|{\tilde{\xi }}_{1}|-{\tilde{\xi }}_{1}^{\left(i\right)}\right)}{\left({\tilde{\xi }}_{1}^{\left(n\right)}-{\tilde{\xi }}_{1}^{\left(i\right)}\right)}\cdot \prod_{m\neq j}\frac{\left(|{\tilde{\xi }}_{2}|-{\tilde{\xi }}_{2}^{\left(j\right)}\right)}{\left({\tilde{\xi }}_{2}^{\left(m\right)}-{\tilde{\xi }}_{2}^{\left(j\right)}\right)}$$
where the new coordinates are ${\tilde{\xi }}_{1}=\xi_{1}-a/2$ and ${\tilde{\xi }}_{2}=\xi_{2}-b/2$, while the control points $\left({\tilde{\xi }}_{1}^{\left(n\right)},{\tilde{\xi }}_{2}^{\left(m\right)}\right)$ are now defined over a grid N×M in $\left[0,a/2\right]\times \left[0,b/2\right]$.

### 2.3. Shell Description

The displacements of an arbitrary point on the shell domain Ω are denoted by $u_{1}$, $u_{2}$ and $u_{3}$. These components correspond to the displacements along the $\xi_{1}$, $\xi_{2}$ and ζ directions, respectively. The component $u_{3}$ is taken positive in the outward direction from the center of the smallest radius of curvature. Considering only von Kàrmàn nonlinearities, the components of the Green–Lagrange strain tensor in the orthogonal curvilinear coordinate system $\left(\xi_{1},\xi_{2},\zeta \right)$ are [45]:(17)$$\begin{array}{cc} \; & \epsilon_{11}=\frac{1}{1+\zeta /R_{1}}\left(\frac{u_{1,1}}{a_{1}}+\frac{a_{1,2}}{a_{1}a_{2}}u_{2}+\frac{u_{3}}{R_{1}}\right)+\frac{1}{2}{\left(\frac{1}{1+\zeta /R_{1}}\right)}^{2}{\left(\frac{u_{3,1}}{a_{1}}\right)}^{2}\\ \; & \epsilon_{22}=\frac{1}{1+\zeta /R_{2}}\left(\frac{u_{2,2}}{a_{2}}+\frac{a_{2,1}}{a_{1}a_{2}}u_{1}+\frac{u_{3}}{R_{2}}\right)+\frac{1}{2}{\left(\frac{1}{1+\zeta /R_{2}}\right)}^{2}{\left(\frac{u_{3,2}}{a_{2}}\right)}^{2}\\ \; & \gamma_{12}=\frac{1}{1+\zeta /R_{1}}\left(\frac{u_{2,1}}{a_{1}}-\frac{a_{1,2}}{a_{1}a_{2}}u_{1}\right)+\frac{1}{1+\zeta /R_{2}}\left(\frac{u_{1,2}}{a_{2}}-\frac{a_{2,1}}{a_{1}a_{2}}u_{2}\right)+\frac{1}{\left(1+\zeta /R_{1}\right)\left(1+\zeta /R_{2}\right)}\left(\frac{u_{3,1}}{a_{1}}\frac{u_{3,2}}{a_{2}}\right)\\ \; & \gamma_{13}=u_{1,\zeta }+\frac{1}{1+\zeta /R_{1}}\left(\frac{u_{3,1}}{a_{1}}-\frac{u_{1}}{R_{1}}\right)\\ \; & \gamma_{23}=u_{2,\zeta }+\frac{1}{1+\zeta /R_{2}}\left(\frac{u_{3,2}}{a_{2}}-\frac{u_{2}}{R_{2}}\right)\\ \; & \epsilon_{33}=u_{3,\zeta } \end{array}$$
where $\epsilon_{\alpha \alpha }$ and $\gamma_{\alpha \beta }$ are the normal and shearing strains, respectively.

The shell kinematics is described according to the first shear deformation theory (FSDT), so:(18)$$\begin{array}{cc} \; & u_{1}\left(\xi_{1},\xi_{2},\zeta ,t\right)=u\left(\xi_{1},\xi_{2}\right)+\zeta \phi_{1}\left(\xi_{1},\xi_{2},t\right)\\ \; & u_{2}\left(\xi_{1},\xi_{2},\zeta ,t\right)=v\left(\xi_{1},\xi_{2}\right)+\zeta \phi_{2}\left(\xi_{1},\xi_{2},t\right)\\ \; & u_{3}\left(\xi_{1},\xi_{2},\zeta ,t\right)=w\left(\xi_{1},\xi_{2}\right) \end{array}$$
where *u*, *v* and *w* are the displacement components of an arbitrary point on Ω along $\xi_{1}$, $\xi_{2}$ and ζ, respectively, while $\phi_{1}$ and $\phi_{2}$ are the rotations. The choice of FSDT is due to the easier FE implementation compared with the classical Kirchhoff theory (CLT). At the same time, FSDT allows shear deformability to be taken into account with a small number of unknown fields. Higher−order theories are not considered here, but can be of practical interest to guarantee improved description of the thickness−wise response, while avoiding the use of the shear factor. The strains of the kinematic model at hand can be obtained upon the substitution of Equation (Equation 18) into Equation (Equation 17):(19)$$\epsilon =\epsilon^{0}+\zeta k,\;\gamma =\gamma^{0}$$
where $\epsilon ={\left\{\epsilon_{11}\;\epsilon_{22}\;\gamma_{12}\right\}}^{T}$ and $\gamma ={\left\{\gamma_{13}\;\gamma_{23}\right\}}^{T}$. The definition of $\epsilon^{0}={\left\{\epsilon_{11}^{0}\;\epsilon_{22}^{0}\;\gamma_{12}^{0}\right\}}^{T}$, $k={\left\{k_{11}\;k_{22}\;k_{12}\right\}}^{T}$ and $\gamma^{0}={\left\{\gamma_{13}^{0}\;\gamma_{23}^{0}\right\}}^{T}$ is available in the Appendix A.

The membrane resultant on the cross section $dA_{2}$, i.e., the section normal to $\xi_{1}$, is:(20)$$\int_{-t/2}^{t/2}\sigma_{11}dA_{2}=\left[\int_{-t/2}^{t/2}\sigma_{11}\left(1+\frac{\zeta }{R_{2}}\right)d\zeta \right]a_{2}d\xi_{2}=N_{11}a_{2}d\xi_{2}$$
where the definition of the resultant $N_{11}$ follows from the equation above. The other resultants are derived in a similar manner, leading to:(21)$$\begin{array}{cc} \left\{\begin{array}{c} N_{11}\\ N_{22}\\ N_{12}\\ N_{21} \end{array}\right\} & =\int_{-t/2}^{t/2}\left\{\begin{array}{c} \sigma_{11}\left(1+\frac{\zeta }{R_{2}}\right)\\ \sigma_{22}\left(1+\frac{\zeta }{R_{1}}\right)\\ \sigma_{12}\left(1+\frac{\zeta }{R_{2}}\right)\\ \sigma_{21}\left(1+\frac{\zeta }{R_{1}}\right) \end{array}\right\}d\zeta ,\;\left\{\begin{array}{c} M_{11}\\ M_{22}\\ M_{12}\\ M_{21} \end{array}\right\}=\int_{-t/2}^{t/2}\zeta \left\{\begin{array}{c} \sigma_{11}\left(1+\frac{\zeta }{R_{2}}\right)\\ \sigma_{22}\left(1+\frac{\zeta }{R_{1}}\right)\\ \sigma_{12}\left(1+\frac{\zeta }{R_{2}}\right)\\ \sigma_{21}\left(1+\frac{\zeta }{R_{1}}\right) \end{array}\right\}d\zeta ,\\ \left\{\begin{array}{c} Q_{1}\\ Q_{2} \end{array}\right\} & =K_{s}\int_{-t/2}^{t/2}\left\{\begin{array}{c} \sigma_{13}\left(1+\frac{\zeta }{R_{2}}\right)\\ \sigma_{23}\left(1+\frac{\zeta }{R_{1}}\right) \end{array}\right\}d\zeta \end{array}$$
where $K_{s}$ is the shear correction factor, which is taken as equal to 5/6. The shear stress resultants, $N_{12}$ and $N_{21}$, and the twisting moments, $M_{12}$ and $M_{21}$, are different due to the curvature terms. For shallow and moderately thick shells, i.e., $h/R_{\alpha }<1/20$, the simplifying assumption $N_{12}\approx N_{21}$ and $M_{12}\approx M_{21}$ can be introduced.

Based on the assumption above, the constitutive law reads:(22)$$\begin{array}{c} \left\{\begin{array}{c} N\\ M\\ Q \end{array}\right\}=\left[\begin{array}{ccc} A(\xi_{1},\xi_{2}) & B(\xi_{1},\xi_{2}) & 0\\ B(\xi_{1},\xi_{2}) & D(\xi_{1},\xi_{2}) & 0\\ 0 & 0 & \bar{A}(\xi_{1},\xi_{2}) \end{array}\right]\left\{\begin{array}{c} \epsilon^{0}\\ k\\ \gamma^{0} \end{array}\right\} \end{array}$$
where $N={\left\{N_{11}\;N_{22}\;N_{12}\right\}}^{T}$, $M={\left\{M_{11}\;M_{22}\;M_{12}\right\}}^{T}$, $Q={\left\{Q_{1}\;Q_{2}\right\}}^{T}$ are the vectors collecting the force and moment resultants, whereas A, D, B and $\bar{A}$ are the membrane, bending, membrane−bending coupling and shear stiffness matrices, respectively.

The equations of motion are derived by referring to Hamilton’s principle:(23)$$\int_{t_{1}}^{t_{2}}\delta \Pi dt=\int_{t_{1}}^{t_{2}}\delta \left[K-\left(U+V\right)\right]dt=0$$
where *K*, *U* and *V* are the kinetic and elastic energies and the potential of the applied loads, respectively. For the shell model developed in this work, these energy quantities are defined as follows: (24)$$\begin{array}{cc} K= & \frac{1}{2}\int_{\Omega }\left[I_{0}\left({\dot{u}}^{2}+{\dot{v}}^{2}+{\dot{w}}^{2}\right)+2I_{1}\left(\dot{\phi_{1}}\dot{u}+\dot{\phi_{2}}\dot{v}\right)+I_{2}\left({\dot{\phi_{1}}}^{2}+{\dot{\phi_{2}}}^{2}\right)\right]a_{1}a_{2}d\xi_{1}d\xi_{2}\\ U= & \frac{1}{2}\int_{\Omega }\left[{\epsilon^{0}}^{T}A\epsilon^{0}+k^{T}Dk+2{\epsilon^{0}}^{T}Bk+{\gamma^{0}}^{T}\bar{A}\gamma^{0}\right]a_{1}a_{2}d\xi_{1}d\xi_{2}\\ V= & -\int_{\Omega }\left[w\left(1+\frac{h}{2R_{1}}\right)\left(1+\frac{h}{2R_{2}}\right)q^{+}\right]a_{1}a_{2}d\xi_{1}d\xi_{2}\\ \; & -\int_{\partial \Omega_{1}}\left(u{\bar{N}}_{11}+v{\bar{N}}_{12}\right)a_{2}d\xi_{2}+\int_{\partial \Omega_{2}}\left(v{\bar{N}}_{22}+u{\bar{N}}_{12}\right)a_{1}d\xi_{1} \end{array}$$
where the dot denotes the time derivative, $q^{+}$ is the pressure applied on the upper surface (ζ=h/2), $\partial \Omega_{1}$ and $\partial \Omega_{2}$ are the boundaries at $\xi_{1}=\mathrm{const}$ and $\xi_{2}=\mathrm{const}$, respectively. The terms $I_{\alpha }$ are the moment of inertia defined as:(25)$$I_{\alpha }=\int_{A}\rho \left(1+\frac{\zeta }{R_{1}}\right)\left(1+\frac{\zeta }{R_{2}}\right){\left(\zeta \right)}^{\alpha }d\zeta \;\alpha =0,1,2$$
where ρ is the mass density.

## 3. The ps−Version of the Finite Element Method

The finite element method (FEM) developed in this work is a combination of two FE schemes, i.e., the p−version of the finite element method (p−FEM) and the s−version (s−FEM). The former is an extension of the conventional FEM, or h−version (h−FEM), where the accuracy is increased by increasing the interpolation order (p−refinement). The latter is an FE scheme where arbitrary local improvements of the mesh resolution are possible through the adoption of advanced mesh superposition techniques (s−refinement). The two features of these numerical approaches are put together into an extended FE framework, called the ps−version of the finite element method (ps−FEM), where simultaneous p− and s−refinement (ps−refinement) can be performed to adaptively adjust the interpolation order *p* and the size *h* of the elements. Within the ps−FEM, the construction of the polynomial space $S^{p}$ is inspired from the p−FEM, while the design of the mesh Δ follows the ideas implemented in the s−FEM, see Figure 2. In the following, the two main pillars of the ps−FEM, i.e., p−FEM and s−FEM, are presented. Then, the ps−FEM is introduced and applied to composite shell problems by referring to the equilibrium equations of Section 2.3.

### 3.1. p−Refinement

The core idea of the *p*−refinement strategy is to progressively increase the polynomial order *p* of the shape functions to improve the quality of the numerical solution. One crucial aspect regards the worsening of the conditionality of the global stiffness matrix as *p* is increased. In the *p*−FEM, this issue can be mitigated by constructing the set of shape functions starting from Legendre polynomials. The properties of these functions enable better numerical conditioning in comparison to the Lagrange polynomials commonly used in the *h*−FEM. High−order polynomials can be considered in the *p*−FEM—even beyond p=3—, without encountering numerical issues.

Considering a generic two−dimensional solution field ϕ depending on *x* and *y*, the FE discretization in the elemental domain $\Omega^{\left(k\right)}$ is:(26)$$\phi^{\left(k\right)}\left(x,y\right)=\phi^{\left(k\right)}\left(\chi^{\left(k\right)}\right)=\sum_{i=1}^{p+1}\sum_{j=1}^{p+1}c_{ij}f_{i}\left(\xi \right)f_{j}\left(\eta \right)\;\mathrm{in}\;\Omega^{\left(k\right)}$$
where −1<{ξ,η}<1 are the nondimensional coordinates in the computational domain $\Omega_{\mathrm{st}}$, $\chi^{\left(k\right)}$ is an array of functions mapping the standard element in the *k*−th element of the mesh Δ, see Figure 3, the coefficients *p* and $c_{ij}$ are the element interpolation order and degrees of freedom, respectively, while $f_{i}$ and $f_{j}$ are polynomial shape functions. In the *p*−FEM the polynomial space $S^{p}$ is constructed as:(27)$$\begin{array}{cc} \; & f_{1}\left(\xi \right)=\frac{1}{2}\left(1+\xi \right),\;f_{2}\left(\xi \right)=\frac{1}{2}\left(1-\xi \right)\\ \; & f_{i+1}\left(\xi \right)=\sqrt{\frac{2i-1}{2}}\int_{-1}^{\xi }P_{i-1}\left(\xi \right)d\xi \;\mathrm{for}\;i=2,3,\ldots p \end{array}$$
where $P_{n}\left(\xi \right)$ is the Legendre polynomial functions of order *n*:(28)$$\begin{array}{cc} \; & P_{0}\left(\xi \right)=1,\;P_{1}\left(\xi \right)=\xi \\ \; & P_{n}\left(\xi \right)=\frac{1}{n}\left[\xi \left(2n-1\right)P_{n-1}\left(\xi \right)-\left(n-1\right)P_{n-2}\left(\xi \right)\right]\;\mathrm{for}\;n=2,3,\ldots p \end{array}$$
From Equation (Equation 27), it is possible to distinguish two families of functions describing different deformation modes, i.e., nodal and internal ones. Nodal modes are represented by $f_{1}\left(\xi \right)$ and $f_{2}\left(\xi \right)$, which are Lagrange linear interpolation polynomials. Internal modes are reproduced by the higher−order terms, $f_{\alpha }\left(\xi \right)$ (for α≥3), which are based on the integrals of Legendre polynomials. The combination of nodal and internal modes leads to different types of two−dimensional shape functions, i.e., nodal (two nodal modes), edge (one nodal mode and one internal mode), and face (two internal modes) shape functions, see Figure 4.

The series expansion in Equation (Equation 27) offers several advantages over the classical Lagrangian polynomial base. The first one is that it inherits the orthogonal properties of the original Legendre polynomials, indeed:(29)$$\int_{-1}^{+1}\frac{\partial f_{i}}{\partial \xi }\frac{\partial f_{j}}{\partial \xi }d\xi =\delta_{ij}\;\mathrm{for}\;i\geq 3,j\geq 1\;\mathrm{or}\;\mathrm{for}\;j\geq 3,i\geq 1$$

This property provides the stiffness matrix with a quasi−diagonal structure, which significantly eases the solution process and enables the use of shape functions with a very high polynomial order. The second feature regards the hierarchical nature of these functions: *p*−refinements implies adding new higher−order terms in Equation (Equation 27), while the low−order ones remain unchanged. As a consequence, the final stiffness matrix is obtained by successively adding rows and columns associated with new levels of *p*−refinements.

Relatively few elements are in most cases sufficient to meet the desired levels of accuracy. At the same time, the method should be capable of appropriately representing the geometry of the structure with a limited number of elements. For this purpose, blending functions [46,47] are usually combined with the *p*−FEM. The mapping procedure for the transformation of the standard element in $\Omega_{\mathrm{st}}$ to the *k*−th element of the mesh in $\Omega^{\left(k\right)}$ is:(30)$$\begin{array}{cc} x=Q_{x}^{\left(k\right)}\left(\xi ,\eta \right)= & f_{1}\left(\eta \right)\left[e_{x1}\left(\xi \right)-f_{1}\left(\xi \right)X_{1}\right]+f_{2}\left(\xi \right)\left[e_{x2}\left(\eta \right)-f_{1}\left(\eta \right)X_{2}\right]\\ \; & f_{2}\left(\eta \right)\left[e_{x3}\left(\xi \right)-f_{2}\left(\xi \right)X_{3}\right]+f_{1}\left(\xi \right)\left[e_{x4}\left(\eta \right)-f_{2}\left(\eta \right)X_{4}\right]\\ y=Q_{y}^{\left(k\right)}\left(\xi ,\eta \right)= & f_{1}\left(\eta \right)\left[e_{y1}\left(\xi \right)-f_{1}\left(\xi \right)Y_{1}\right]+f_{2}\left(\xi \right)\left[e_{y2}\left(\eta \right)-f_{1}\left(\eta \right)Y_{2}\right]\\ \; & f_{2}\left(\eta \right)\left[e_{y3}\left(\xi \right)-f_{2}\left(\xi \right)Y_{3}\right]+f_{1}\left(\xi \right)\left[e_{y4}\left(\eta \right)-f_{2}\left(\eta \right)Y_{4}\right] \end{array}$$
where $Q_{x}^{\left(k\right)}\left(\xi ,\eta \right)$ and $Q_{y}^{\left(k\right)}\left(\xi ,\eta \right)$ are general nonlinear mapping functions, $\left(X_{\alpha },Y_{\alpha }\right)$ are the nodal coordinates of the element, while $e_{\alpha x}$ and $e_{\alpha y}$ with α=1,…,4 are the functions defining the curves of the element edge, as illustrated in Figure 3.

From the mapping of Equation (Equation 30), the derivatives with respect to the physical coordinates *x* and *y* are computed as:(31)$$\left\{\begin{array}{c} \frac{\partial }{\partial x}\\ \frac{\partial }{\partial y} \end{array}\right\}={\left[\begin{array}{cc} \frac{\partial Q_{x}^{\left(k\right)}}{\partial \xi } & \frac{\partial Q_{y}^{\left(k\right)}}{\partial \xi }\\ \frac{\partial Q_{x}^{\left(k\right)}}{\partial \eta } & \frac{\partial Q_{y}^{\left(k\right)}}{\partial \eta } \end{array}\right]}^{-1}\left\{\begin{array}{c} \frac{\partial }{\partial \xi }\\ \frac{\partial }{\partial \eta } \end{array}\right\}={\left[\begin{array}{cc} J_{11}^{\left(k\right)} & J_{12}^{\left(k\right)}\\ J_{21}^{\left(k\right)} & J_{22}^{\left(k\right)} \end{array}\right]}^{-1}\left\{\begin{array}{c} \frac{\partial }{\partial \xi }\\ \frac{\partial }{\partial \eta } \end{array}\right\}$$
where $J_{\alpha \beta }^{\left(k\right)}$ are the components of the Jacobian matrix $J^{\left(k\right)}$.

Line integrals at the four sides of the element read:(32)$$\begin{array}{cc} \; & \int_{e_{\alpha }}F\left(x,y\right)dx=\int_{-1}^{1}F\left(\xi ,\pm 1\right)\sqrt{{J_{11}^{\left(k\right)}}^{2}+{J_{12}^{\left(k\right)}}^{2}}d\xi \;\mathrm{for}\;\alpha =1,3\\ \; & \int_{e_{\alpha }}F\left(x,y\right)dy=\int_{-1}^{1}F\left(\pm 1,\eta \right)\sqrt{{J_{21}^{\left(k\right)}}^{2}+{J_{22}^{\left(k\right)}}^{2}}d\eta \;\mathrm{for}\;\alpha =2,4 \end{array}$$
while the surface ones are:(33)$$\int_{\Omega^{\left(k\right)}}F\left(x,y\right)dxdy=\int_{-1}^{1}\int_{-1}^{1}F\left(\xi ,\eta \right)Jd\xi d\eta$$
where *F* is the generic integrand, F is obtained from *F* by replacing *x* and *y* with the mapping functions in Equation (Equation 30), while $J=\det J^{\left(k\right)}$ is the determinant of the Jacobian matrix.

### 3.2. *s*−Refinement

The main problem in performing local mesh refinements relates to the need to generate a transition between refined and unrefined regions. Different approaches are available in the literature to address this issue, for example transition elements or multi−point constraints approaches [24,25,26,27]. The *s*−refinement strategy offers the advantage of simplifying this process by allowing an element size reduction in the desired regions only. This result is achieved through the definition of an independent local/fine mesh which is superimposed to a global/coarse one [30].

Using this idea, the final FE approximation ϕ can be represented as:(34)$$\phi =\left\{\begin{array}{c} \phi_{G}\;\mathrm{in}\;\Omega -\Omega_{L}\\ \phi_{G}+\phi_{L}\;\mathrm{in}\;\Omega_{L} \end{array}\right.$$
where $\phi_{G}$ is the global mesh solution defined in Ω, while $\phi_{L}$ is the local mesh solution defined in $\Omega_{L}\subset \Omega$, see Figure 5. Note that the mesh superposition technique allows for incompatible discretization between global and local mesh. This gives an extremely high level of flexibility when performing local *h*−refinements, as no transition regions [24,25,26] or multi-point constraints [27] are required.

This concept of solution superposition can be extended to multi−level refinements [41]. In this case, the elements of the local mesh can be further refined by superposing one over the other multiple levels of overlaid meshes. In this case, the final FE solution ϕ becomes:(35)$$\phi =\left\{\begin{array}{c} \phi_{G}\;\mathrm{in}\;\Omega -\Omega_{L}^{\left(1\right)}\\ \phi_{G}+\phi_{L}^{\left(1\right)}\;\mathrm{in}\;\Omega_{L}^{\left(1\right)}-\Omega_{L}^{\left(2\right)}\\ \ldots \\ \phi_{G}+\phi_{L}^{\left(1\right)}+\ldots +\phi_{L}^{\left(s\right)}+\ldots +\phi_{L}^{\left(N_{s}\right)}\;\mathrm{in}\;\Omega_{L}^{\left(N_{s}\right)}\; \end{array}\right.$$
where $\phi_{L}^{\left(s\right)}$ is the local solution given by the mesh at level *s* covering the domain $\Omega_{L}^{\left(s\right)}\subset \Omega_{L}^{\left(s-1\right)}\subset \ldots \subset \Omega$. Note that the solution on the global $\phi_{G}$ and local meshes $\phi_{L}^{\left(s\right)}$ can be represented by any FE scheme, such as the *p*−FEM presented in Section 3.1.

Two conditions, i.e., compatibility of the basis functions and their linear independency, are required to apply this multi−level decomposition of the solution field ϕ.

The first condition implies $C^{1}$−continuity within each elements and $C^{0}$−continuity across the element boundaries. The $C^{1}$−continuity is satisfied by construction. On the contrary, the inter−element continuity is not guaranteed and needs to the imposed. This is achieved by enforcing homogeneous Dirichlet boundary conditions on the boundary of the overlaid meshes, as depicted in Figure 6a.

The second condition on the linear independency is required to avoid singularities in the stiffness matrix. In general, the redundant degrees−of−freedom can be removed during the factorization process by elimination of the equations with zero pivots [30]. If the *p*−FEM is employed for the global and local meshes, this is avoided by ensuring that shape functions of the same type (nodal, side, face) and polynomial order *p* appear only once in regions with multiple meshes. This idea is graphically illustrated in Figure 6b.

### 3.3. ps−Refinement

The ps−refinement procedure is a combination of the *p*− and *s*−FEM approaches. In this framework, the order of interpolation *p* can be increased with no numerical issues owing to the properties of Legendre polynomials; at the same time, the elements size *h* can be adaptively reduced by overlaying different levels of superposition meshes *s*.

In this work, the ps−refinement is used to develop an advanced FEM, called the ps−FEM, which is exploited to perform global/local analysis of laminated shells. In particular, the *p*−refinement strategy is exploited to capture the smooth features of the solution at global scale, such as the global deformation field, while the *s*−refinement approach is employed to capture local effects, such as stress concentrations.

The refinement process is outlined in Figure 7. First, a global/base mesh $\Delta_{G}$ is defined. The resolution of $\Delta_{G}$ is chosen to guarantee appropriate definition of the loading and boundary conditions, and to avoid the description of the geometry using distorted elements. Then, a *p*−refinement is performed until the required accuracy on the global response is reached. Finally, the solution is *s*−refined with local meshes in the regions where enhanced interpolation capability is required.

The ps−FEM shares similar convergence features with the hp−FEM. Indeed, by increasing the order of the polynomial expansion in combination with overlaid meshes, it is possible achieve an exponential decaying global approximation error. This is true even when the solution presents steep gradients or singular points. More insights on convergence properties can be found in the work of [42].

In the proposed framework, three different combinations of *p*− and *s*−refinement are allowed, namely linear, uniform, and graded ps−refinements. In the linear case [39], the global solution is represented by a coarse high−order mesh, while locally a multi−level *s*−refinement is performed with low−order meshes. The uniform and graded ps−refinement strategies [41] are an extension of the previous one, where high−order local meshes are employed. In the first case, the local meshes have the same order *p* of the global one, while in the second case *p* is different from level to level.

In principle, *s*−refinement can be carried out considering local meshes with arbitrary resolution and orientation, as originally done in [30]. A simplified approach is implemented here, inspired by the work of [41]. In particular, the elements of the refined region are divided to half of the original size. These elements are then used to define the superposed mesh at the upper level. This procedure allows for a quicker refinement, with no need to iterate to correlate the generic coordinates of the local and global element domain [30].

In the ps−FEM, the shell generalized displacements for the *k*−th element with interpolation order *p* at the *s*−level mesh are:(36)$$\begin{array}{cc} u(\xi_{1},\xi_{2},t) & =u\left(\chi^{\left(k,s\right)},t\right)=\sum_{p=1}^{p_{1}+1}\sum_{q=1}^{p_{1}+1}c_{pq}^{\left(1\right)}\left(t\right)f_{p}\left(\xi \right)f_{q}\left(\eta \right)\\ v(\xi_{1},\xi_{2},t) & =v\left(\chi^{\left(k,s\right)},t\right)=\sum_{r=1}^{p_{2}+1}\sum_{s=1}^{p_{2}+1}c_{rs}^{\left(2\right)}\left(t\right)f_{r}\left(\xi \right)f_{s}\left(\eta \right)\\ w(\xi_{1},\xi_{2},t) & =w\left(\chi^{\left(k,s\right)},t\right)=\sum_{m=1}^{p_{3}+1}\sum_{n=1}^{p_{3}+1}c_{mn}^{\left(3\right)}\left(t\right)f_{m}\left(\xi \right)f_{n}\left(\eta \right)\\ \phi_{1}\left(\xi_{1},\xi_{2},t\right) & =\phi_{1}\left(\chi^{\left(k,s\right)},t\right)=\sum_{i=1}^{p_{4}+1}\sum_{j=1}^{p_{4}+1}c_{ij}^{\left(4\right)}\left(t\right)f_{i}\left(\xi \right)f_{j}\left(\eta \right)\\ \phi_{2}\left(\xi_{1},\xi_{2},t\right) & =\phi_{2}\left(\chi^{\left(k,s\right)},t\right)=\sum_{k=1}^{p_{5}+1}\sum_{l=1}^{p_{5}+1}c_{kl}^{\left(5\right)}\left(t\right)f_{k}\left(\xi \right)f_{l}\left(\eta \right)\;\mathrm{in}\;\Omega^{\left(k\right)} \end{array}$$
where $p_{\alpha }$ and $c^{\left(\alpha \right)}$ (α=1,2,3,4,5) are the element expansion order and unknown amplitude coefficients for the displacement fields, while $\chi^{\left(k,s\right)}$ is the vector collecting the mapping functions between the element computational space $\Omega_{\mathrm{st}}^{\left(k,s\right)}$ and element physical space $\Omega^{\left(k,s\right)}$. This mapping procedure is illustrated in Figure 8. For the global mesh elements (*s* = 0), the vector of mapping functions is defined by Equation (Equation 30). For the superimposed mesh elements (s>0), the following sequence of mapping is operated:(37)$$\begin{array}{c} \chi^{\left(k,s\right)}=\chi^{\left(k,0\right)}\circ \Psi^{\left(k,1\right)}\circ \;...\;\circ \Psi^{\left(k,s-1\right)}\circ \Psi^{\left(k,s\right)} \end{array}$$
where $\Psi^{\left(k,s\right)}$ is the vector collecting the mapping functions between the computational domain of the element and the underlying one, as shown in Figure 8. Each element is therefore provided with two set of mapping functions, the global one $\chi^{\left(k,s\right)}$ and the local one $\Psi^{\left(k,s\right)}$. These functions are required for defining the derivatives and integrating at element level.

It is noted that the ps−FEM can be applied for the numerical solution of any mathematical model, such as higher−order shear deformation theories, as well as three−dimensional elasticity theories. The effectiveness of FE techniques relying on mesh superposition has been discussed in the literature for such problems. Examples can be found in [42], where the hp−*d*−FEM is employed to solve 3D linear elastodynamic problems, and in [32,35] where the *s*−FEM is employed in the context of multiple model methods.

Using the approximation of Equation (Equation 36), the energy contributions of Equation (Equation 24) are:(38)$$\begin{array}{cc} K^{\left(k,s\right)}= & \frac{1}{2}[M_{\left(pq)(\bar{\mathrm{pq}}\right)}^{\left(11\right)}{\dot{c}}_{pq}^{\left(1\right)}{\dot{c}}_{\bar{\mathrm{pq}}}^{\left(1\right)}+M_{\left(rs)(\bar{\mathrm{rs}}\right)}^{\left(22\right)}{\dot{c}}_{rs}^{\left(2\right)}{\dot{c}}_{\bar{\mathrm{rs}}}^{\left(2\right)}+M_{\left(mn)(\bar{\mathrm{mn}}\right)}^{\left(33\right)}{\dot{c}}_{mn}^{\left(3\right)}{\dot{c}}_{\bar{\mathrm{mn}}}^{\left(3\right)}+M_{\left(ij)(\bar{\mathrm{ij}}\right)}^{\left(44\right)}{\dot{c}}_{ij}^{\left(4\right)}{\dot{c}}_{\bar{\mathrm{ij}}}^{\left(4\right)}+M_{\left(kl)(\bar{\mathrm{kl}}\right)}^{\left(55\right)}{\dot{c}}_{kl}^{\left(5\right)}{\dot{c}}_{\bar{\mathrm{kl}}}^{\left(5\right)}\\ + & 2M_{\left(pq)(mn\right)}^{\left(14\right)}{\dot{c}}_{pq}^{\left(1\right)}{\dot{c}}_{mn}^{\left(4\right)}+2M_{\left(pq)(mn\right)}^{\left(25\right)}{\dot{c}}_{pq}^{\left(2\right)}{\dot{c}}_{mn}^{\left(5\right)}]\\ U^{\left(k,s\right)}= & \frac{1}{2}[K_{\left(pq)(\bar{\mathrm{pq}}\right)}^{\left(11\right)}c_{pq}^{\left(1\right)}c_{\bar{\mathrm{pq}}}^{\left(1\right)}+2K_{\left(pq)(rs\right)}^{\left(12\right)}c_{pq}^{\left(1\right)}c_{rs}^{\left(2\right)}+2K_{\left(pq)(mn\right)}^{\left(13\right)}c_{pq}^{\left(1\right)}c_{mn}^{\left(3\right)}+2K_{\left(pq)(ij\right)}^{\left(14\right)}c_{pq}^{\left(1\right)}c_{ij}^{\left(4\right)}+2K_{\left(pq)(kl\right)}^{\left(15\right)}c_{pq}^{\left(1\right)}c_{kl}^{\left(5\right)}\\ + & K_{\left(rs)(\bar{\mathrm{rs}}\right)}^{\left(22\right)}c_{rs}^{\left(2\right)}c_{\bar{\mathrm{rs}}}^{\left(2\right)}+2K_{\left(rs)(mn\right)}^{\left(23\right)}c_{rs}^{\left(2\right)}c_{mn}^{\left(3\right)}+2K_{\left(rs)(ij\right)}^{\left(24\right)}c_{rs}^{\left(2\right)}c_{ij}^{\left(4\right)}+2K_{\left(rs)(kl\right)}^{\left(25\right)}c_{rs}^{\left(2\right)}c_{kl}^{\left(5\right)}\\ + & K_{\left(mn)(\bar{\mathrm{mn}}\right)}^{\left(33\right)}c_{mn}^{\left(3\right)}c_{\bar{\mathrm{mn}}}^{\left(3\right)}+2K_{\left(mn)(ij\right)}^{\left(34\right)}c_{pq}^{\left(3\right)}c_{ij}^{\left(4\right)}+2K_{\left(mn)(kl\right)}^{\left(35\right)}c_{pq}^{\left(3\right)}c_{kl}^{\left(5\right)}\\ + & K_{\left(ij)(\bar{\mathrm{ij}}\right)}^{\left(44\right)}c_{ij}^{\left(4\right)}c_{\bar{\mathrm{ij}}}^{\left(4\right)}+2K_{\left(ij)(kl\right)}^{\left(45\right)}c_{ij}^{\left(4\right)}c_{kl}^{\left(5\right)}\\ + & K_{\left(kl)(\bar{\mathrm{kl}}\right)}^{\left(55\right)}c_{kl}^{\left(5\right)}c_{\bar{\mathrm{kl}}}^{\left(5\right)}\\ + & N_{\left(pq)(mn)(\bar{\mathrm{mn}}\right)}^{\left(133\right)}c_{pq}^{\left(1\right)}c_{mn}^{\left(3\right)}c_{\bar{\mathrm{mn}}}^{\left(3\right)}+N_{\left(rs)(mn)(\bar{\mathrm{mn}}\right)}^{\left(233\right)}c_{rs}^{\left(2\right)}c_{mn}^{\left(3\right)}c_{\bar{\mathrm{mn}}}^{\left(3\right)}+N_{\left(ij)(mn)(\bar{\mathrm{mn}}\right)}^{\left(433\right)}c_{ij}^{\left(4\right)}c_{mn}^{\left(3\right)}c_{\bar{\mathrm{mn}}}^{\left(3\right)}\\ + & N_{\left(kl)(mn)(\bar{\mathrm{mn}}\right)}^{\left(533\right)}c_{kl}^{\left(5\right)}c_{mn}^{\left(3\right)}c_{\bar{\mathrm{mn}}}^{\left(3\right)}+N_{\left(mn\right)\left(\bar{\mathrm{mn}}\right)\left(\tilde{mn}\right)}^{\left(333\right)}c_{mn}^{\left(3\right)}c_{\bar{\mathrm{mn}}}^{\left(3\right)}c_{\tilde{mn}}^{\left(3\right)}+N_{\left(mn\right)\left(\bar{\mathrm{mn}}\right)\left(\tilde{mn}\right)\left(\hat{mn}\right)}^{\left(3333\right)}c_{mn}^{\left(3\right)}c_{\bar{\mathrm{mn}}}^{\left(3\right)}c_{\tilde{mn}}^{\left(3\right)}c_{\hat{mn}}^{\left(3\right)}]\\ V^{\left(k,s\right)}= & P_{pq}^{\left(1\right)}c_{pq}^{\left(1\right)}+P_{rs}^{\left(2\right)}c_{rs}^{\left(2\right)}+P_{mn}^{\left(3\right)}c_{mn}^{\left(3\right)} \end{array}$$
where use is made of the index notation presented in [48]. According to this notation, $a_{pq}$ represents a vector, $A_{\left(pq)(rs\right)}$, $A_{\left(pq)(rs)(mn\right)}$, $A_{\left(pq)(rs)(mn)(ij\right)}$ are second−, third− and fourth−order fourth−order arrays.

By application of Hamilton’s principle to the generic element, the following kernel is obtained:(39)$$\int_{t_{1}}^{t_{2}}\delta \Pi^{\left(k,s\right)}dt=\delta {c^{\left(k,s\right)}}^{T}M^{\left(k,s\right)}{\ddot{c}}^{\left(k,s\right)}+\delta {c^{\left(k,s\right)}}^{T}K^{\left(k,s\right)}c^{\left(k,s\right)}+\delta c^{\left(k,s\right)}P^{\left(k,s\right)}=0$$
where $u^{\left(k,s\right)}$ is the vector collecting the element degrees of freedom, $M^{\left(k,s\right)}$, $K^{\left(k,s\right)}$, $P^{\left(k,s\right)}$ are the element stiffness, mass matrices, and load vector.

Upon assembly of the contributions of all elements, the governing equations are obtained as follows:(40)$$M\ddot{c}+Kc+N_{2}c+N_{3}c=P$$
where M, K and P are the global mass, linear stiffness matrices and vector of external loads, respectively, while $N_{2}$ and $N_{3}$ are the nonlinear contributions due to cubic and quartic terms in the elastic energy, respectively. The set of ODEs reported in Equation (Equation 40) describe the nonlinear dynamics of the shell and can be solved via implicit or explicit integration techniques. In this work, the discussion is restricted to the quasi−static case, so the relevant equations simplify to:(41)$$Kc+N_{2}c+N_{3}c=P$$
for which the solution is computed by referring to the Newton–Raphson method.

## 4. Results

This section presents applications of ps−FEM to laminated shell problems. The method has been implemented in a Matlab environment, while all analysis have been run on a laptop with the following characteristics: 1.4 GHz Intel Core i7 processor, 16 GB 1867 MHz LPDDR3 memory, Intel HD Graphics 6,151,536 MB.

In the first part of the section, a series of validation studies is proposed. Then, the numerical technique proposed herein is exploited to solve exemplary test cases for laminated shell problems. The potential of the ps−refinement is illustrated against classical *h*− and *p*−refinements for linear and geometrically nonlinear problems.

### 4.1. Validation

#### 4.1.1. Test Case 1: Vibrations of Elliptical Shells

The first study aims at validating the shell model implemented and, for this purpose, the vibration response is studied for three elliptical cylindrical shells. The test case is taken from the literature [49] and covers the case of shells characterized by different eccentricities *e*. The position vector of the reference surface is expressed as:(42)$$r\left(\xi_{1},\xi_{2}\right)=\xi_{1}e_{x}+R_{A}\sin\left(\xi_{2}\right)e_{y}+R_{B}\cos\left(\xi_{2}\right)e_{z}\;\mathrm{for}\;\xi_{1}\in \left[0,L\right]\;\mathrm{and}\;\xi_{2}\in \left[0,2\pi \right]$$
where $R_{A}$ and $R_{B}$ are the semi−major and semi−minor axis, respectively, while *L* is the length. A sketch of the shell is provided in Figure 9, while the geometric configurations considered in this study are summarized in Table 1. The shells are made of an aluminum whose properties are E=68,950 MPa, ν=0.3, and ρ=2766 kgm/${}^{3}$. Clamped−free boundary conditions are considered.

The numerical model consists of a coarse mesh with five elements along the circumferential direction and two along the axial one, see Figure 10. The polynomial order is p=10.

The results are reported in Table 2 for the first ten natural frequencies. A comparison is illustrated against the reference results taken from [49], where the same shell kinematics and geometric formulation were adopted. In [49] the shell governing equations are expressed in strong−form and are solved using the generalized differential quadrature method. Excellent matching is achieved for the reported frequencies, where the error is evaluated as $E_{\%}=\left|\omega -\omega_{\mathrm{ref}}\right|/\omega_{\mathrm{ref}}\times 100$. For the reported modes, the percentage errors are of the order of $E_{\%}=10^{-2}-10^{-3}$. The almost perfect agreement gives evidence of the correct implementation of the shell geometry and kinematics.

#### 4.1.2. Test Case 2: Vibration of Variable Stiffness Plate

The second study aims at validating the ability of the proposed numerical code to handle the case of structures with non−uniform stiffness. In this regards, the example presented herein provides a validation of the correctness of the constitutive law implemented [50].

A free vibration analysis is conducted by considering a variable stiffness (VS) rectangular laminate, as illustrated in Figure 11. In particular, the geometry is defined by a=b=300 mm and t=1.2 mm. The material is characterized by the following elastic properties: $E_{11}=$ 10,000 MPa, $E_{22}=9000$ MPa, $G_{12}=G_{13}=G_{23}=5000$ MPa, $\nu_{12}=\nu_{13}=\nu_{23}=0.3$. The stacking sequence is described in compact manner as ${\left[\pm T\right]}_{2s}$, where the entries of the the matrix T are:(43)$$T=\langle T_{11}|T_{12}\rangle \;\mathrm{with}\;T_{11}=-45\;\mathrm{and}\;T_{12}=\left\{45,30,15,0,-15,-30,-45\right\}$$
where different values of $T_{12}$ define different fiber paths. Starting from the configurations defined by Equation (Equation 43), the local fiber orientation is obtained by referring to Equation (Equation 16).

The VS plate is clamped at the four edges and is modeled with one single element of polynomial order *p*. The results are presented in terms of the first nondimensional frequency $\Omega =\omega a\sqrt{\rho /E_{22}}$ in Table 3. The comparison against the reference results is provided by considering different degrees of *p*−refinement. As shown, the FE results converge quickly upon *p*−refinement with frequencies converging from above. This behavior is motivated by the hierarchical properties of the Legendre expansion [6]. It is important to note that different FE discretizations may imply different distributions of the elastic properties leading to nonmonothonic convergence of the solution. The comparison of the converged frequencies with the reference ones [50] provides differences below 1%, thus illustrating the possibility of successfully handling structures with non−uniform elastic properties within the present framework.

#### 4.1.3. Test Case 3: Static Analysis of Plate with Cutout

This test case deals with the static analysis of plate with circular cutout loaded in traction. The presence of the cutout determines the onset of stress concentrations, thus providing a useful mean for investigating the validity of the proposed ps−refinement strategies.

The plate, an illustration of which is reported in Figure 12, is characterized by dimensions a=75 mm, b=50 mm and r=10 mm. The elastic properties considered for the material are: $E_{11}=$ 150,000 MPa, $E_{22}=9000$ MPa, $G_{12}=G_{13}=G_{23}=5000$ MPa, $\nu_{12}=\nu_{13}=\nu_{23}=0.32$. A quasi−isotropic layup ${\left[\pm 45,90,0\right]}_{s}$ is considered and each ply has thickness $t_{\mathrm{ply}}=1$ mm. The traction load is prescribed at two parallel edges with magnitude ${\bar{N}}_{xx}=100$ N/mm. The FE model is restricted to a quarter of the full structure owing to the double symmetry of the problem.

The *s*−refinement strategy is considered for designing the mesh of the numerical model. In particular, a base mesh having five elements with polynomial order *p* is superposed with s=5 levels of overlaying meshes of decreasing size, as illustrated in Figure 13. In the validation, the order *p* is increased until convergence of the global solution is reached. The number of overlaying layers *s* is selected to have a proper representation of the local response, while the resolution of the base mesh (number/size of elements) is chosen to guarantee the appropriate subdivision of the domain for performing *s*−refinements in the regions of interest. Specifically, the overlaying meshes are placed around the corners of the cutout, where stress gradients are expected.

Three FE models employing the ps−refinement strategies illustrated in Figure 7 are set up for the validation study. The interpolation order of the superimposing elements are $p_{s}=1$ and $p_{s}=p$ for the linear and uniform ps−refinement, respectively. Regarding the graded ps−refinement, the polynomial order $p_{s}$ is set according to the following law:(44)$$p_{s}=p-\mathrm{floor}\left(\frac{s}{m}\right)$$
where *m* is a constant that controls the rate of decrease of *p* for increasing levels *s*. The law of Equation (Equation 44) will be adopted in the rest of this work for the graded ps−refinement strategy, unless otherwise specified. In the present study, the ps−refinements are performed fixing the mesh and increasing the polynomial degree of the elements on the global mesh *p*, while the order of the overlaying elements $p_{s}$ is adjusted according to the specific refinement strategy, i.e., linear, uniform, or graded.

The validation is carried out using a FE model realized in Abaqus with S4 shell elements. A preliminary convergence study was carried out with this model. Based on this study, the total number of elements has been taken equal to 350,000 elements (*h*−refinement) to guarantee convergence. The corresponding number of membrane degrees of freedom is $7\times 10^{5}$, approximately. Results are shown in Table 4 for different types and levels of ps−refinements, and are presented in terms of nondimensional elastic energy:(45)$$\bar{U}=U\frac{E_{11}}{{\bar{N}}_{xx}^{2}r^{2}t^{3}}$$

From Table 4, it can be seen that the nondimensional elastic energies of the three FE models approach the reference one as *p* is increased. In particular, one can note a quick monotonic convergence from below, i.e., the strain energy increases as the FE models are more refined. This is an expected behaviour as the numerical model becomes more flexible with an increasing number of degrees of freedom. For the linear and graded ps−refinement strategies, convergence is reached with p=5. Regarding the uniform ps−refinement, convergence is achieved with a slightly lower polynomial order (p=4). This is due to the use of high−order elements in the upper mesh levels which enable us to obtain a higher approximation capability.

In Figure 14, the convergence curves are presented for the stress resultant $N_{xx}$ measured at $\left(\xi_{1},\xi_{2}\right)=\left(0,r\right)$ for the three FE models considered for the validation. In all cases, one can note a strong and non−monotonic convergence of $N_{xx}$ to the reference value of 378.26 MPa. The curves in Figure 14 provide insights on the local convergence rate of the ps−refinement strategies. For the present problem, convergence of the stress measures is reached for approximately $10^{3}$ degrees of freedom, which is much lower than the one required by the reference model where a *h*−refinement was employed.

The consistency of these results, both in terms of global and local quantities, provide a validation for the ps−refinement strategies implemented in this work.

### 4.2. Applications

#### 4.2.1. Example 1: Vibration and Buckling of a Highly Anisotropic Laminated Plate

The first application of the ps−FEM is an example taken from [51], and relates to the vibration and buckling analysis of a highly anisotropic composite plate. The following data are considered for the analysis: a/b=1 and a/t= 100,000, see Figure 11. The material has elastic properties given by: $E_{11}=$ 393,000 MPa, $E_{22}=5030$ MPa, $G_{12}=G_{13}=G_{23}=5240$ MPa, $\nu_{12}=\nu_{13}=\nu_{23}=0.31$, ρ=1500 kg/$m^{3}$. The plate is a laminate with a single ply oriented at θ=45. Simply supported boundary conditions are considered along the four edges, while uniform edge−shortening conditions are considered for the buckling analysis.

This problem has been studied in previous efforts in the literature due to its challenging convergence features. The eigenvalues of the problem, both for free vibrations and buckling, tend to convergence with a slow rate as a consequence of the high orthotropy ratio $E_{11}/E_{22}=73.36$, whose effects are exacerbated by the 45−oriented ply. Indeed, this orientation promotes strong bending/twisting elastic couplings with highly localized gradients in the modal shapes.

Two meshes are realized for conducting the numerical tests. The first one is a mesh with 5 × 5 quadrilateral elements, see Figure 15a. The second one is a *s*−refined mesh, whose features are outlined in Figure 15b; the *s*−refinement is obtained starting from the base mesh 5 × 5 and adding 15 layers of overlaying meshes. The refinement is conducted to allow local effects arising from vibration and buckling modes to be accurately captured.

Starting from these two meshes, five FE models are developed to illustrate the effect of different refinement strategies. The first two models, *h*− and *p*−models, employ the mesh in Figure 15a and implement the *h*− and *p*−refinement strategies, respectively. The other three models, ps−L−, ps−U−, and ps−G−models, are constructed from the *s*−refined mesh in Figure 15b and adopt the linear (L), uniform (U), and graded (G) ps−refinement strategies, respectively. A summary of the FE models is reported in Table 5, whose nomenclature will be used for the other examples reported next.

In absence of available exact solutions, the first step consists of obtaining highly accurate results to be used as a reference for comparing the five models later. The ps−U−model is used for this scope. The results are summarized in Table 6 in terms of nondimensional frequency and buckling load, defined as:(46)$$\hat{\omega }=\omega \frac{a^{2}}{t}\sqrt{\frac{\rho }{E_{22}}},\;{\hat{N}}_{xx}={\bar{N}}_{xx}\frac{a^{2}}{E_{22}t^{3}}$$
where ${\bar{N}}_{xx}$ is the force per unit length on the loaded edges.

As shown, a value of p=7 guarantees convergence up to the third digit for both vibration and buckling parameters. These values are retained as references for computing the errors obtained with different refinement strategies.

The vibration and buckling mode shapes are illustrated in Figure 16 and Figure 17. Both modes are characterized by one single−skew half−wave, where stretching occurs in the direction of fibers, i.e., θ=45. From Figure 16a and Figure 17a, one can note that the deflected shape *w* is smooth in most of the domain apart from the two corners. Here, it is possible to note highly localized effects, as evident from the plots of the twisting moments $M_{xy}$ in Figure 16b and Figure 17b where strong stress concentrations are observed. This will have drastic consequences on the convergence of the fundamental frequency and critical buckling load.

The relative errors percentage from the numerical models of Table 5 are summarized in Figure 18, where an additional curve in the results of [51], i.e., Ritz *p*−refinement, is reported. The plots report the logarithmic relative errors percentage evaluated against the reference solutions.

Despite quantitative distinctions, similar trends are observed for the frequency and the buckling results, the latter requiring more degrees of freedom for a fixed corresponding error. From Figure 18, the fast convergence rate achieved with the uniform and graded ps−refinements is clearly visible. In particular, the ps−U− and ps−G−refinements outperform FE models implementing standard *h*− and *p*−refinement approaches. This effect is clear from the error−to−degrees−of−freedom ratio of the nondimensional frequency and buckling load in Figure 18, with the ps−U− and ps−G−models presenting the curves with the steepest slopes. The excellent convergence properties just shown are made possible by the flexibility of the approach presented here: the degrees of freedom can be employed to refine specific areas of the domain with no need of generating transition regions, where refinement is not useful in terms of solution accuracy, but is a mean to generate compatible meshes.

The plots of Figure 18 allow us to distinguish regions where the adoption of a refinement strategy is advantageous over the others. For instance, percentage errors $E_{\%}$ of the order of 0.1–1% suggest the adoption of a *p*−refinement strategy. On the contrary, if stricter requirements are set, the ps−refinements provide the best mean for solving the problem, with uniform and graded strategies as the most effective ones.


*Benchmark results*


The previous section demonstrated the advantages of the ps−FEM in deriving highly accurate solutions with relatively low computational costs. The proposed ps−FEM framework is then exploited to provide reference results to be used for comparison purposes in future studies in the field.

The same plate considered earlier are studied, but the investigation is now extended to different orientation angles, θ=[30,45,60], and orthotropic ratios, $E_{11}/E_{22}=\left[73.36,40,20,10\right]$. By combining these values, twelve plates with different levels of anisotropy are obtained.

The analyses are performed using the ps−U−model considered in Table 6. The mesh is illustrated in Figure 15b, and the corresponding total number of degrees of freedom is approximately $2\times 10^{4}$.

A summary of the results is provided in Table 7 in terms of the nondimensional eigenvalues according to Equation (Equation 46). The results are compared with the predictions obtained using the Ritz method proposed in [51]. Specifically, the Ritz predictions are obtained by expanding the unknowns with 250 × 250 Ritz functions, corresponding to a total of $6\times 10^{4}$ degrees of freedom, i.e., six times higher than the present model. As shown, the eigenvalues obtained via ps−U−model are always smaller than the Ritz ones. Owing to the convergence properties of the solution (the eigenvalues converge from above) the present results are thus more accurate. The maximum differences are reached for the plate with $E_{11}/E_{22}=73.36$ and θ=45, which corresponds to the largest degree of anisotropy. Note, for the buckling load, the difference is as high as 0.40%, approximately, with respect to an already refined solution. The difference between the present and reference results becomes smaller as the degree of anisotropy is reduced either by reducing the off−axis angle or the orthotropy ratio.

#### 4.2.2. Example 2: Stress Analysis of Variable Stiffness Elliptical Shell

The response of a cylindrical laminated shell is investigated under the effect of a static point load. The shell has an elliptical cross−section with semi−axes $R_{A}=1000$ mm and $R_{B}=500$ mm, total thickness t=10 mm, and length L=2000 mm. The sketch of the structure is reported in Figure 9. The material has elastic properties $E_{11}=$ 150,000 MPa, $E_{22}=9000$ MPa, $G_{12}=G_{13}=G_{23}=5000$ MPa, $\nu_{12}=\nu_{13}=\nu_{23}=0.32$, while the stacking sequence is given as:(47)$${\left[\pm T\right]}_{s}\;\mathrm{with}\;T=\left[30,45,30\right]$$
where the fiber path is obtained via Equation (Equation 15). The shell is clamped at both ends and is loaded with a transverse point load with magnitude P=100 N and applied at $\left(\xi_{1},\xi_{2}\right)=\left(L/2,3/2\pi \right)$. This loading condition is of interest due to the localized phenomena associated with the concentrated force and finds practical application in tests such as the single perturbation load analysis (SPLA) [52], used for the assessment of shell imperfection sensitivity.

Two FE models are developed and compared. The two models share the same number of degrees of freedom, $10^{4}$, approximately, but have different features for the meshing strategy. The first model relies upon a *p*−refinement strategy (*p*−model) and is presented in Figure 19a. Its mesh has 8 × 8 elements, i.e., four elements along the circumferential and the axial directions, with a polynomial order p=11.

The second model adopts the *s*−refined mesh illustrated in Figure 19b. The base mesh is unaltered with respect to the *p*−model. A local refinement is introduced with s=5 layers of overlaying meshes in the surroundings of the loaded region. The uniform ps−refinement strategy (ps−U−model) is considered, with the polynomial order of the base elements set to p=5.

The plots of the out−of−plane displacement *w* and the stress component $\sigma_{22}$ on the outmost ply at $\xi_{2}=3/2\pi$ are illustrated in Figure 20. Similar results are obtained for the two models in terms of deflections, as shown in Figure 20a.

The advantages of the ps−refinement are clear when addressing the predicted stress $\sigma_{22}$ in Figure 20b, where the discrepancies between the two models are noticeable. An excellent prediction of the stress gradient, which is particularly exacerbated close in the loaded region, is achieved via ps−refinement. The comparison between the peak values of the two models reveals an underprediction of the *p*−model that is close to 60%.

The contour of the stress is reported in Figure 21. As shown, the *p*−model solution presents severe oscillations in the surroundings of the stress peak (Figure 21a). This phenomenon is typically observed in high−order approximations when the solution displays steep gradients or discontinuities. Oscillations are greatly attenuated in the case of the ps−refined solution, as revealed by Figure 21b. These results provide a clear insight into the advantages of the proposed strategy. By recalling that the two models rely upon the same number of unknowns, evidence is given on the superiority of the ps−refinement strategy for problem characterized by strong gradients. As a matter of fact, the combination of *p*− and *s*−refinements allows us to combine the advantages of both strategies. The ability of high−order approximations is exploited to capture effectively the global response, while the superposed mesh provides an embedded approach suitable for detecting local effects.

#### 4.2.3. Example 3: Snap−Back of Cylindrical Panel

The static nonlinear response of a shell is discussed in this closing example. The test case is taken from [53] and considers an isotropic cylindrical panel loaded by a point force applied at its center and directed inward. The panel is illustrated in Figure 22, where the following parameters are considered: a=b=50.8 mm, R=254 mm, t=0.635 mm. An isotropic material is considered with Young modulus E=310.125 MPa and Poisson ratio ν=0.3. The panel is under simply−supported boundary conditions along the straight edges, $\xi_{2}=0,b$, while free conditions are imposed along the curved ones, $\xi_{1}=0,a$.

The results are presented by comparing the *h*−model, *p*−model and ps−L−model of Table 5. In all cases, the symmetry of the problem is exploited by modeling one quarter of the structure and imposing symmetry boundary conditions.

A convergence study is performed by considering two levels of refinement for each model, as displayed in Figure 23. The *h*−refinement procedure is performed for a fixed polynomial order p=1, with increased mesh resolution, as reported in Figure 23a. The *p*−refinement strategy is carried out by increasing the polynomial order, p=1,3,6, on a mesh of 2 × 2 element, see Figure 23b. The ps−refinement is performed by increasing both the polynomial order *p* and the degree of *s*−refinement of the mesh *s*. More specifically, the *p*− and *s*−orders are increased simultaneously with steps of one (p=1,2,3) and five (s=0,5,10), respectively, see Figure 23c. For clarity, a summary of the FE refinement parameters is presented in Table 8 along with the number of degrees of freedom.

The results are presented in terms of load−deflection curves in Figure 24. An arc−length continuation procedure has been implemented for capturing the snap−back. As shown, all the solutions converge to the reference one provided sufficient refinement steps are performed. In particular, the results of Figure 24a demonstrate that 2 steps of refinement, leading to a total of 1327 dofs, are required for the *h*−model. On the contrary, one single step of refinement is needed for the *p*−model and the ps−L−model, as clear from the plots of Figure 24b,c. In these cases, the number of degrees of freedom is one order of magnitude lower for the *p*−model (343) and ps−L−model (252). These results demonstrate the advantage implied by the use of appropriate meshing and refinement strategies. Through a comparison of the *p*− and ps−refinement strategies with a classical *h*−refinement approach, the quality of the predictions is preserved, but the number of degrees of freedom can be as low as 1/5. This saving becomes even more important in the context of nonlinear analyses, where the overall process requires repeated matrix factorizations and linear systems to be solved.

## 5. Conclusions

This paper dealt with the application of the ps−version of the finite element method to the analysis of composite thin shells. The approach relies on the possibility of superposing multiple local meshes in arbitrary regions to refine the solution only where actually needed. No transition meshes are required nor special−purpose elements to connect areas with different grid densities.

The implementation proposed herein allows a wide class of structures to be analyzed, including shells with arbitrary curvature and innovative materials with variable in−plane elastic properties. Both geometrically linear and nonlinear features are introduced. Three different combinations of *p*− and *s*−refinements were considered, although other strategies could be easily developed and implemented.

The proposed advanced refinement technique is useful under several circumstances in the analysis of composite shells. The results illustrate this aspect for the case of local stress concentrations, where the possibility of superposing multiple layers of refinement can be exploited to achieve accurate stress predictions with reduced effort. For 2D laminated structures, advanced refinement can be of interest even in case of global responses, such as for the eigenanalysis of highly anisotropic plates. Indeed, owing to complex elastic couplings, the ability to capture local effects can have a drastic influence also on the predictions at a global level. For these problems, conventional methods based on high−order global approximations can be unsuitable if refined predictions are of concern. Among the different combinations of *p*− and *s*−refinements considered in the various test cases, no optimal choice can be established ex ante. The choice is, in general, problem−dependent.

The inherently hierarchical nature of the proposed tool suggests its use at different stages of the design process of composite shells. In the early design phases, simplified models with one or few macro elements with high−order interpolation can be the most suitable way for obtaining quick estimates. In more advanced phases, when increased detail is needed, the models can be easily improved by exploiting the local refinement capabilities to maximize the ratio between accuracy and number of degrees of freedom.

## Figures and Tables

**Figure 1 materials-16-01395-f001:**
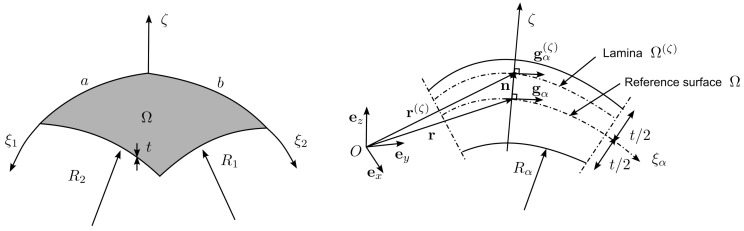
Shell geometry definition.

**Figure 2 materials-16-01395-f002:**
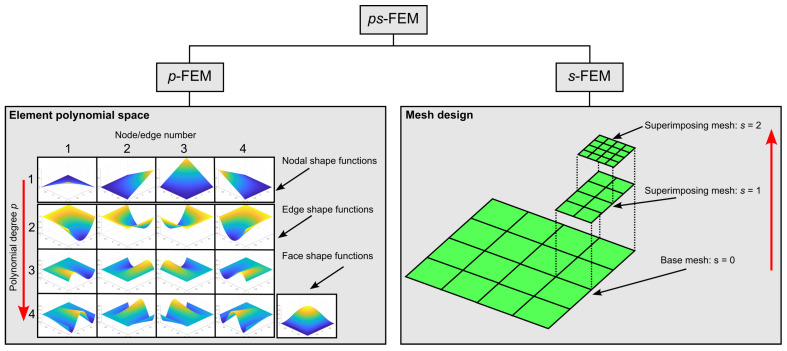
Building blocks of the ps−FEM framework.

**Figure 3 materials-16-01395-f003:**
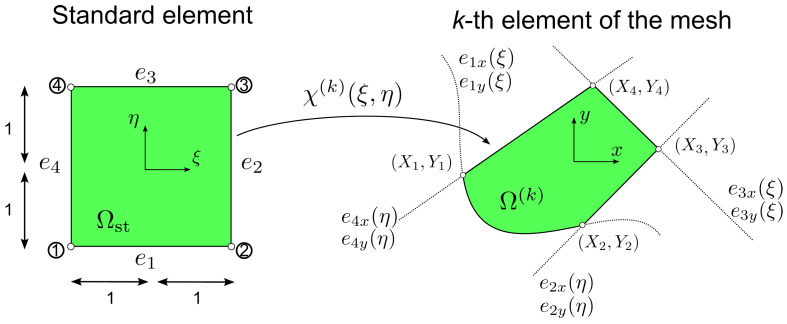
Element mapping procedure.

**Figure 4 materials-16-01395-f004:**
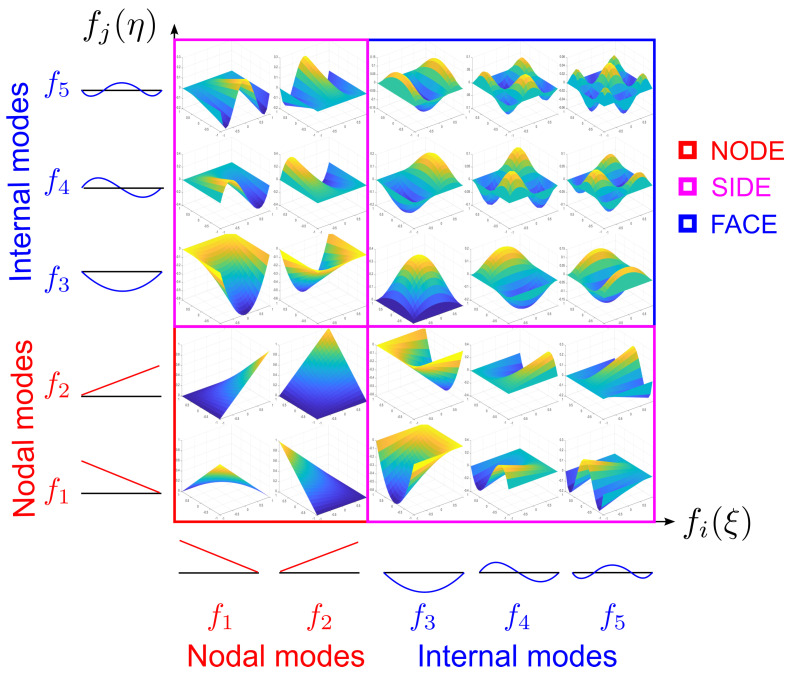
Two−dimensional shape functions.

**Figure 5 materials-16-01395-f005:**
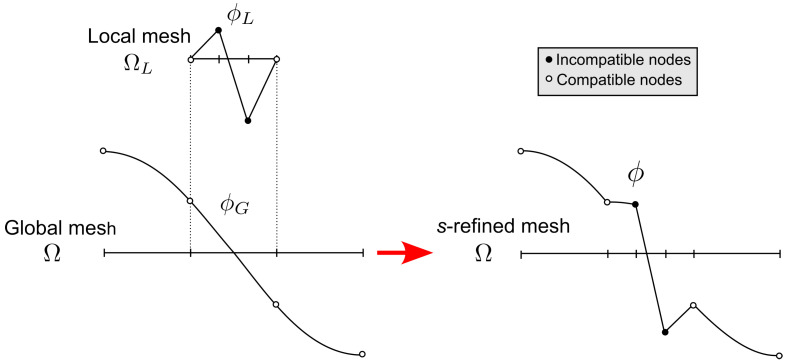
Conceptual idea of mesh superposition.

**Figure 6 materials-16-01395-f006:**
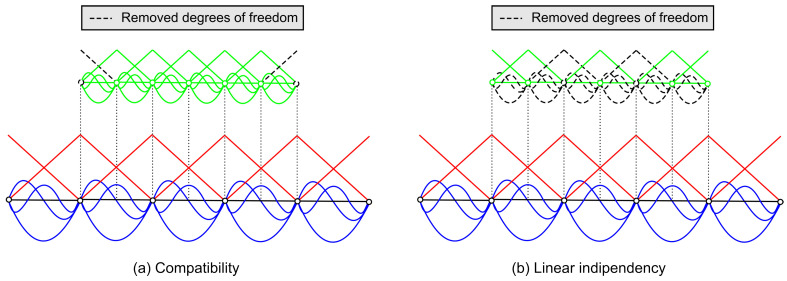
Conditions to satisfy when performing *s*−refinements: (**a**) compatibility and (**b**) linear indipendency.

**Figure 7 materials-16-01395-f007:**
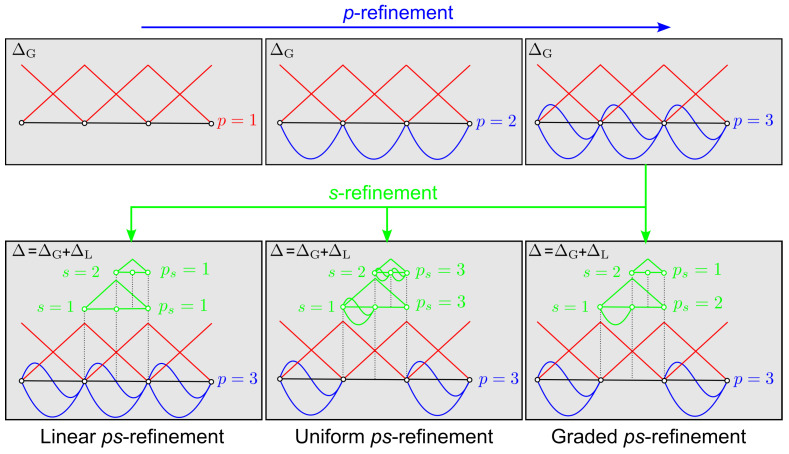
Refinement process in the ps−FEM.

**Figure 8 materials-16-01395-f008:**
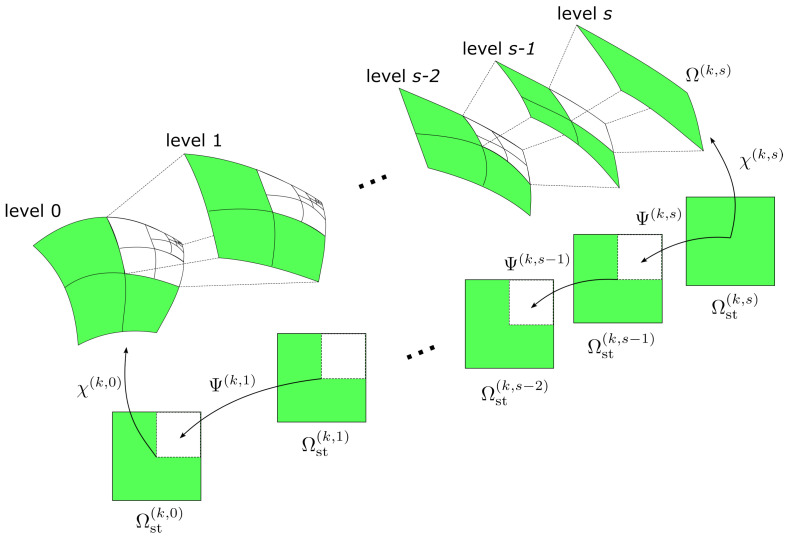
Mapping procedure between local meshes and global one.

**Figure 9 materials-16-01395-f009:**
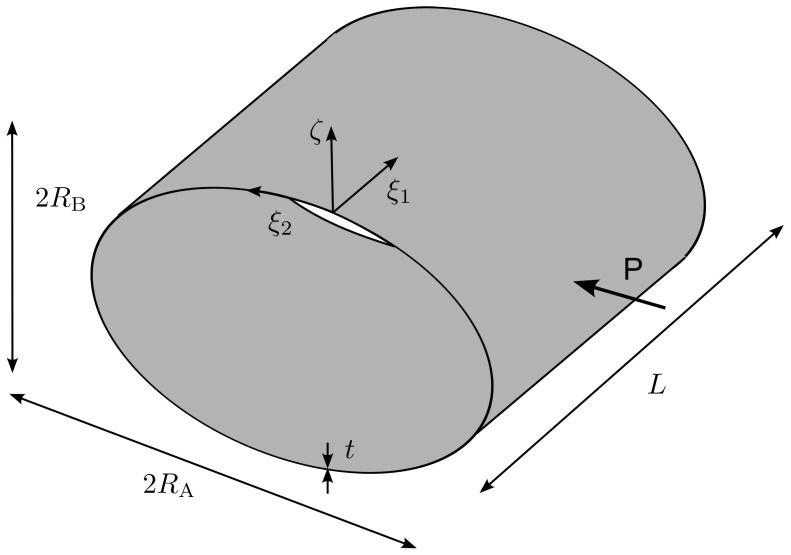
Cylindrical elliptical shell: geometry.

**Figure 10 materials-16-01395-f010:**
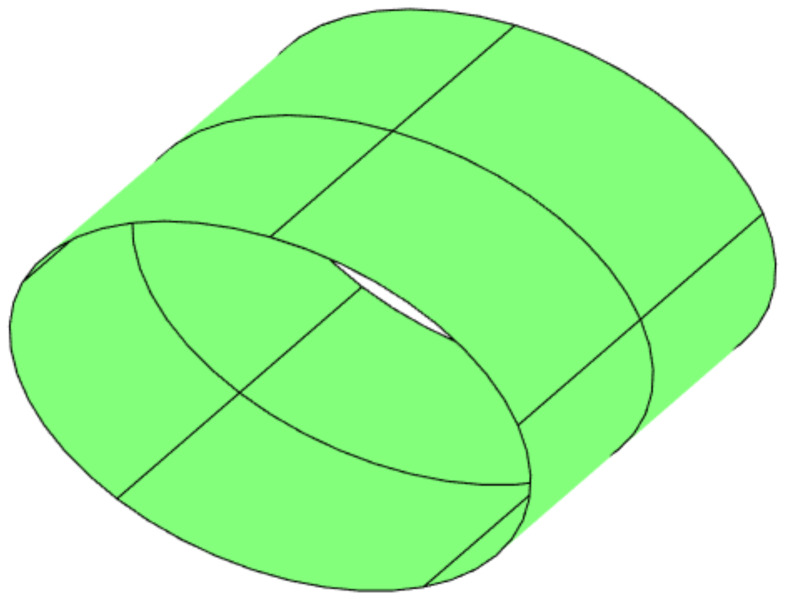
Cylindrical elliptical shell: mesh.

**Figure 11 materials-16-01395-f011:**
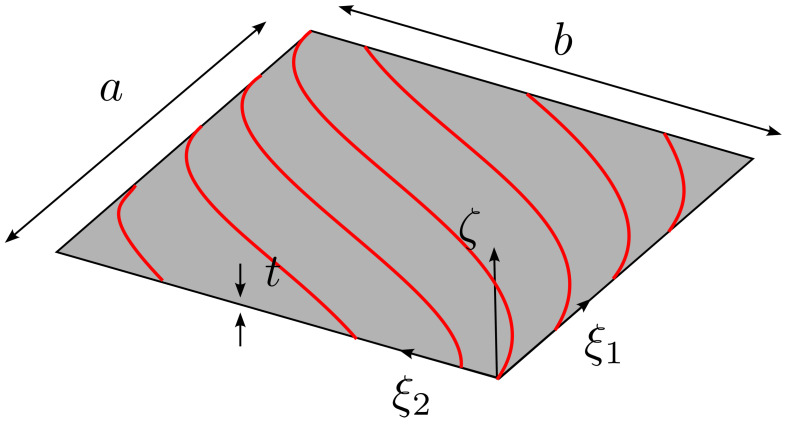
Variable stiffness plate—Geometry.

**Figure 12 materials-16-01395-f012:**
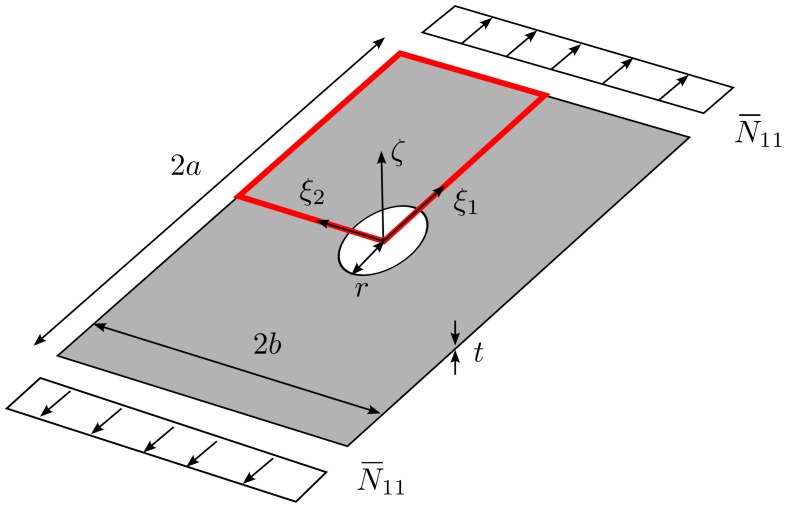
Plate with cutout: geometry.

**Figure 13 materials-16-01395-f013:**
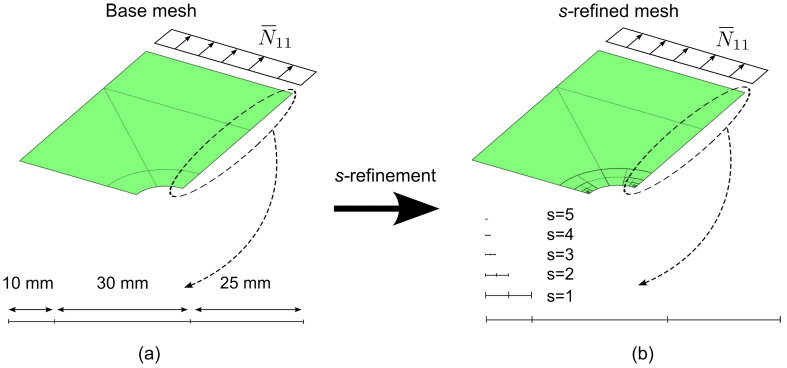
Plate with cutout — static analysis: (**a**) Base mesh, (**b**) s−refined mesh.

**Figure 14 materials-16-01395-f014:**
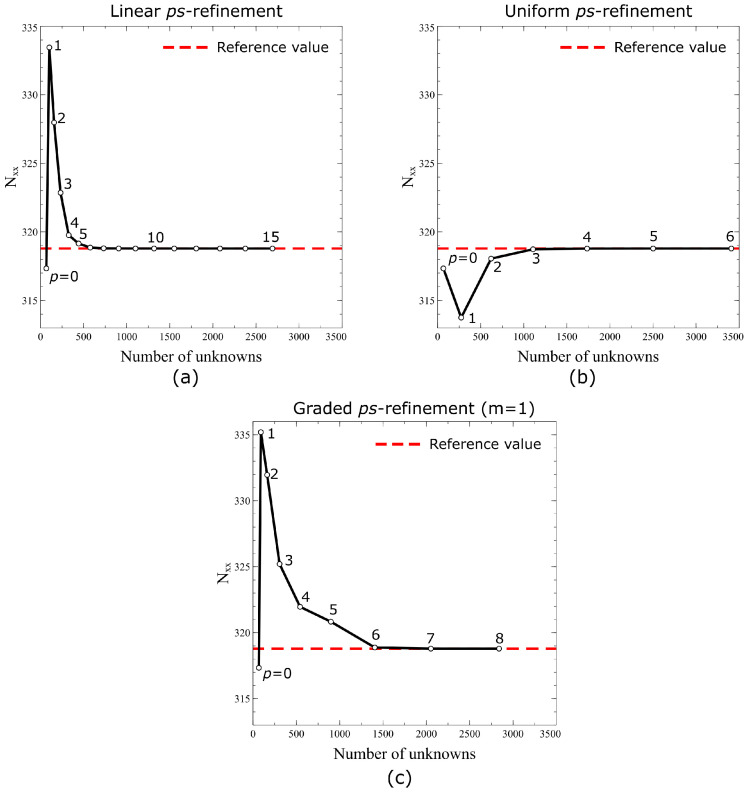
Computed values of $N_{xx}$ at $\left(\xi_{1},\xi_{2}\right)=\left(0,r\right)$ for different levels and types of ps−refinement: (**a**) linear, (**b**) uniform, and (**c**) graded.

**Figure 15 materials-16-01395-f015:**
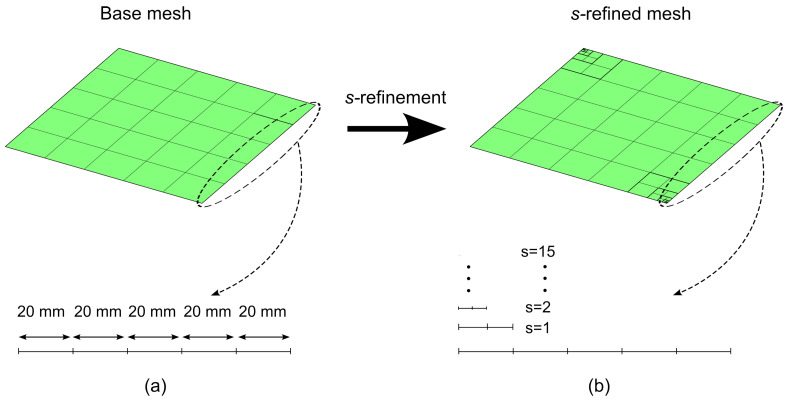
Highlyanisotropic plate – vibration and buckling analysis: (**a**) Base mesh, (**b**) s−refined mesh.

**Figure 16 materials-16-01395-f016:**
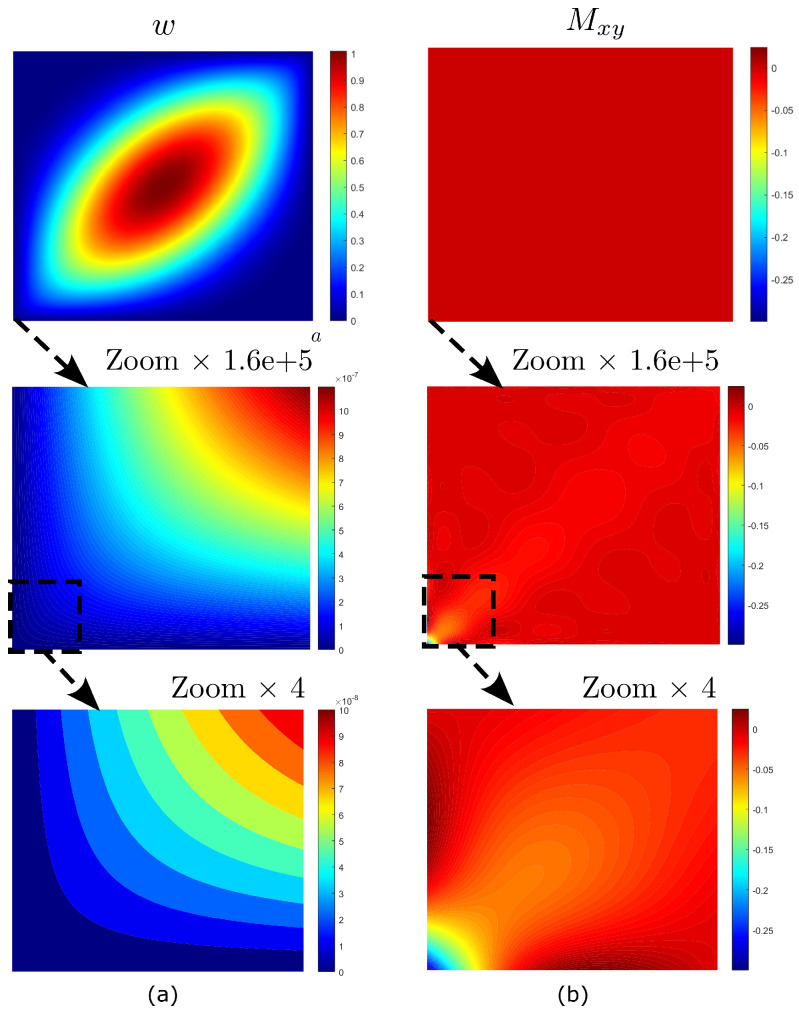
Highly anisotropic plate: mode shapes for free vibration analysis: (**a**) deflection shape and (**b**) twisting moment.

**Figure 17 materials-16-01395-f017:**
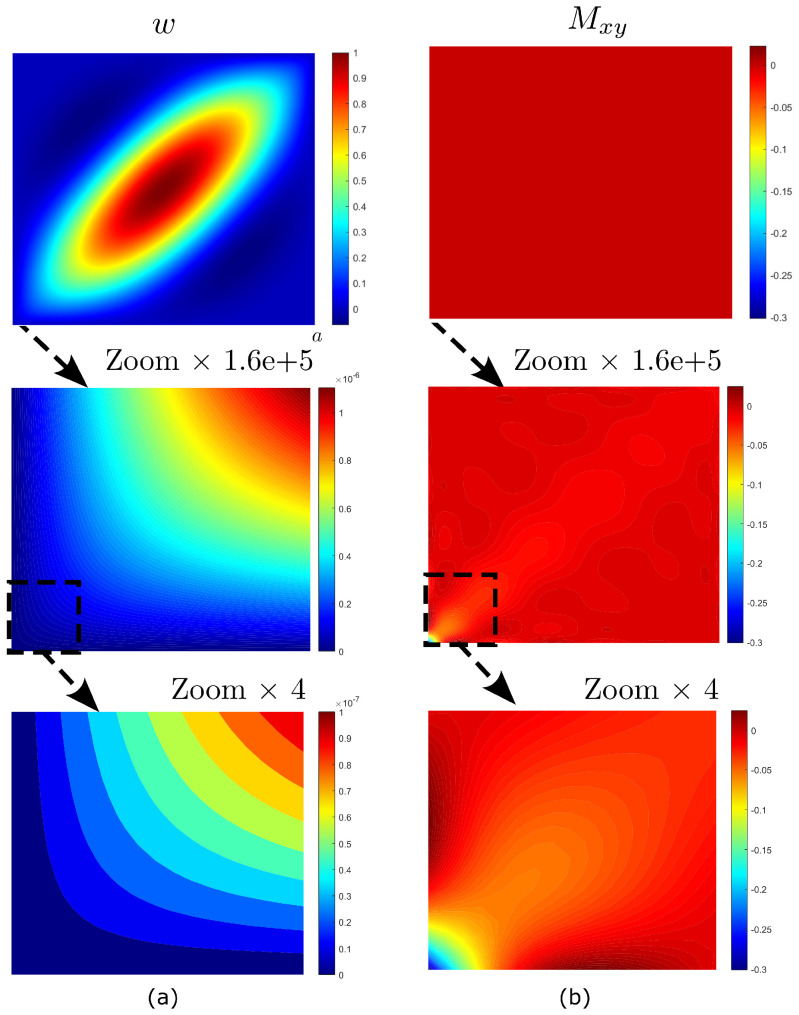
Highly anisotropic plate: mode shapes for buckling analysis: (**a**) deflection shape and (**b**) twisting moment.

**Figure 18 materials-16-01395-f018:**
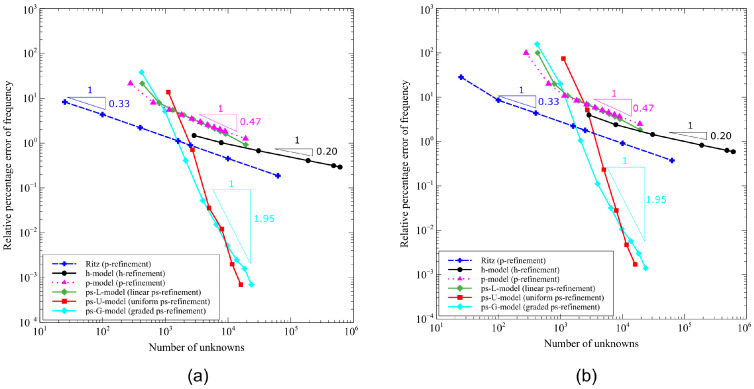
Highly anisotropic plate: convergence for different refinement strategies: (**a**) nondimensional fundamental frequency and (**b**) nondimensional buckling load.

**Figure 19 materials-16-01395-f019:**
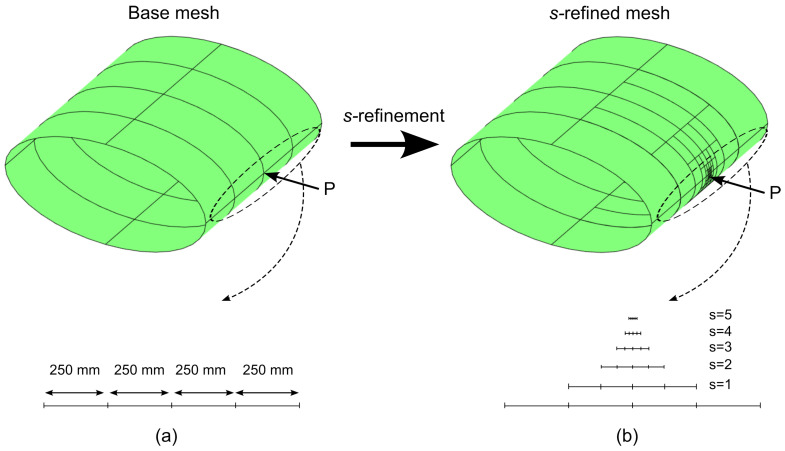
Variable Stiffness cylindrical elliptical shell – static analysis: (**a**) Base mesh, (**b**) s−refined mesh.

**Figure 20 materials-16-01395-f020:**
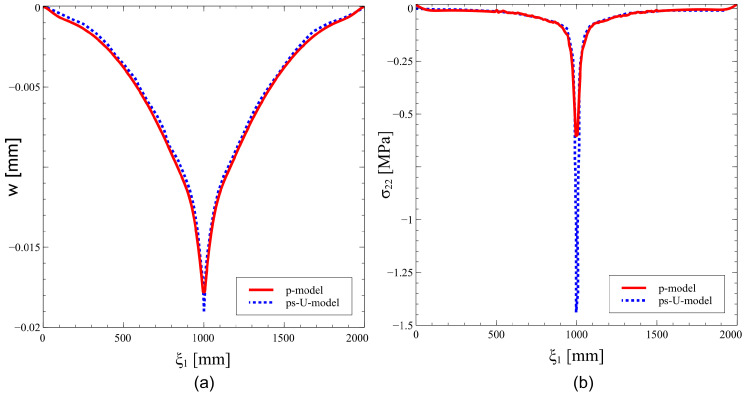
Variable stiffness cylindrical elliptical shell: comparison between *p*− and ps−U−models: (**a**) displacement and (**b**) normal stress field.

**Figure 21 materials-16-01395-f021:**
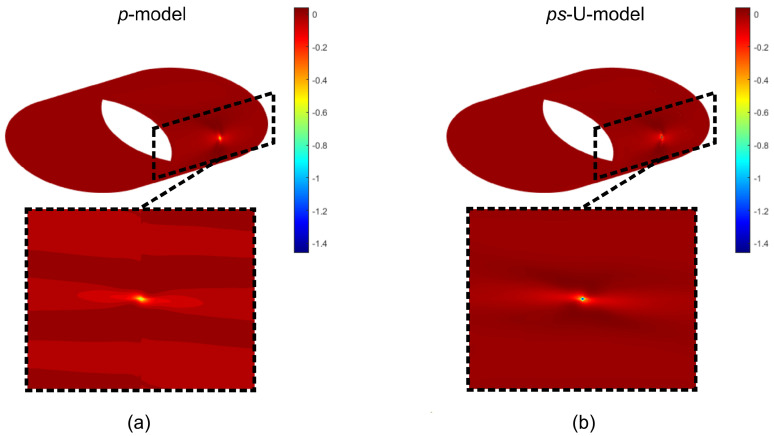
Variable stiffness cylindrical elliptical shell: stress field $\sigma_{22}$: (**a**) *p*−model and (**b**) ps−U−model.

**Figure 22 materials-16-01395-f022:**
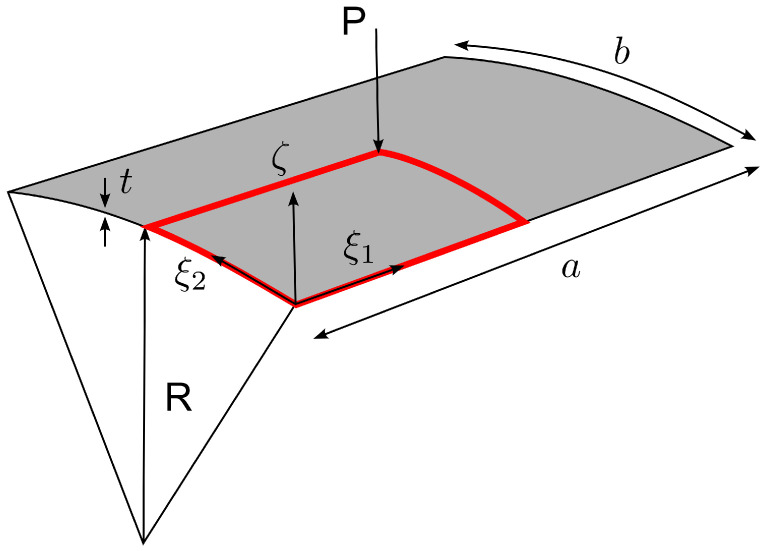
Geometry of cylindrical panel.

**Figure 23 materials-16-01395-f023:**
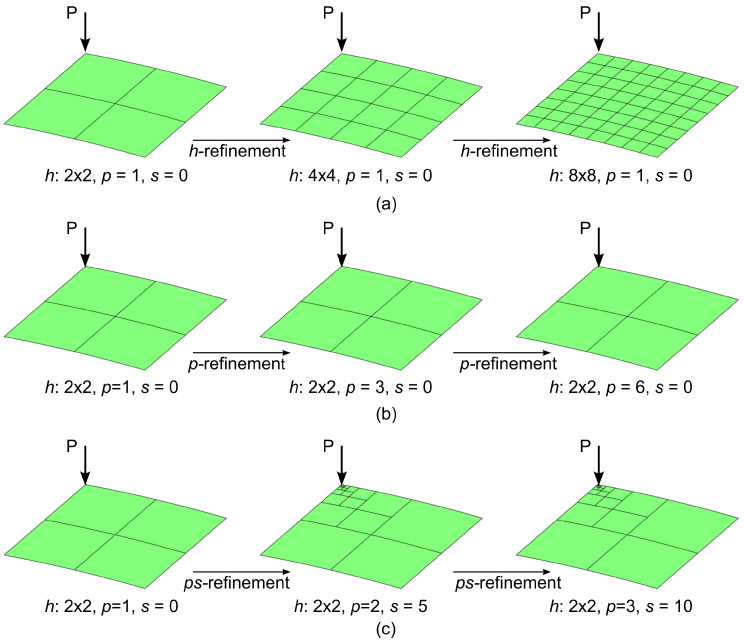
Cylindrical panel: refinement strategies: (**a**) *h*, (**b**) *p*, (**c**) ps−linear.

**Figure 24 materials-16-01395-f024:**
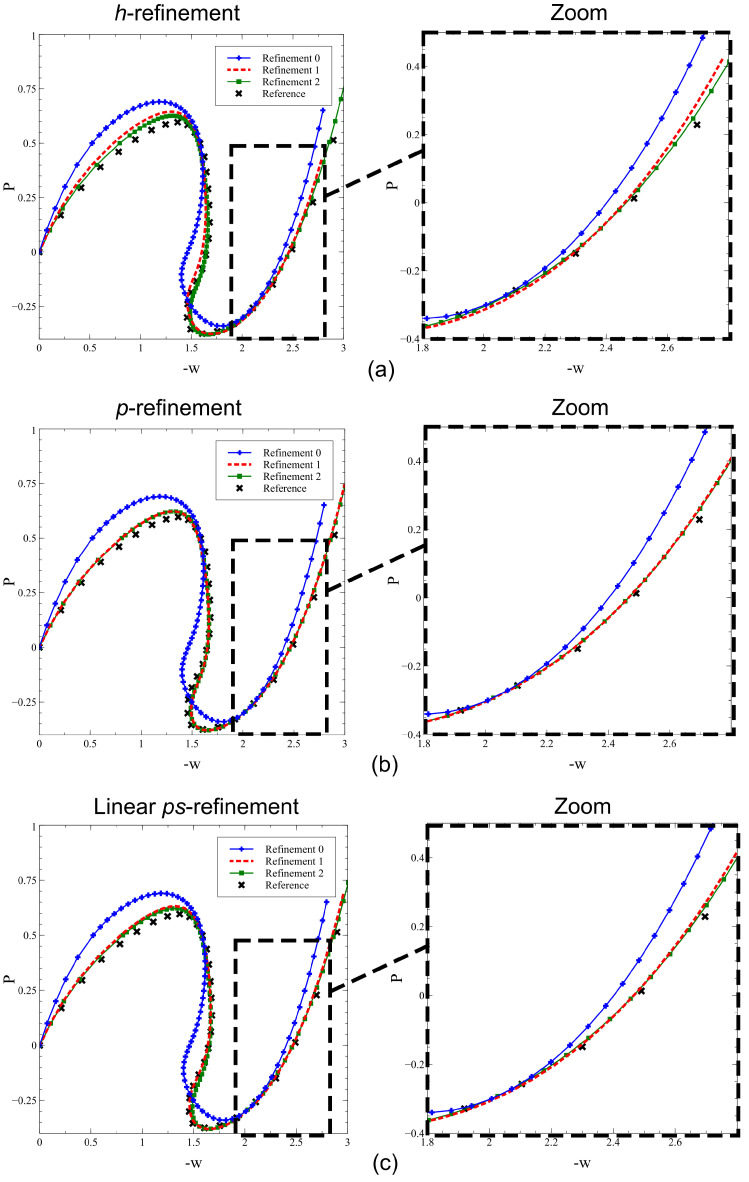
Cylindrical panel: load−deflection curves for different levels of (**a**) *h*−, (**b**) *p*− and (**c**) ps−refinements.

**Table 1 materials-16-01395-t001:** Shell geometry.

	*L* (mm)	$R_{A}$ (mm)	$R_{B}$ (mm)	*e* (-)
Shell 1	594.9188	304.8000	304.8000	0.0000
Shell 2	595.3250	328.9300	279.6540	0.5265
Shell 3	595.3760	365.5060	237.4900	0.7601

**Table 2 materials-16-01395-t002:** Natural frequencies ω (rad/s) for elliptical shells with different eccentricities *e*. Subscript: percent difference against ref. [49].

Mode N°	e=0.0000	e=0.5265	e=0.7601
1	611.2500${}^{\left(0.0031\right)}$	597.7019${}^{\left(0.0033\right)}$	539.5629${}^{\left(0.0041\right)}$
2	611.2500${}^{\left(0.0031\right)}$	597.7020${}^{\left(0.0034\right)}$	539.5683${}^{\left(0.0059\right)}$
3	643.1510${}^{\left(0.0003\right)}$	618.4085${}^{\left(0.0016\right)}$	542.8651${}^{\left(0.0025\right)}$
4	643.1510${}^{\left(0.0003\right)}$	618.4086${}^{\left(0.0015\right)}$	542.8681${}^{\left(0.0023\right)}$
5	701.4954${}^{\left(0.0042\right)}$	712.9803${}^{\left(0.0025\right)}$	715.4793${}^{\left(0.0009\right)}$
6	701.4954${}^{\left(0.0042\right)}$	712.9810${}^{\left(0.0025\right)}$	715.5105${}^{\left(0.0049\right)}$
7	857.9875${}^{\left(0.0014\right)}$	847.2782${}^{\left(0.0010\right)}$	785.3787${}^{\left(0.0042\right)}$
8	857.9875${}^{\left(0.0014\right)}$	847.2840${}^{\left(0.0014\right)}$	785.5405${}^{\left(0.0080\right)}$
9	864.4224${}^{\left(0.0025\right)}$	870.6317${}^{\left(0.0001\right)}$	892.8773${}^{\left(0.0072\right)}$
10	864.4224${}^{\left(0.0025\right)}$	870.6408${}^{\left(0.0003\right)}$	893.0180${}^{\left(0.0147\right)}$

**Table 3 materials-16-01395-t003:** Natural frequencies (rad/s) of rectangular plate with different variable stiffness layups. Subscript: percent difference against ref. [50].

*p*	$\langle -45|45\rangle$	$\langle -45|30\rangle$	$\langle -45|15\rangle$	$\langle -45|0\rangle$	$\langle -45|-15\rangle$	$\langle -45|-30\rangle$	$\langle -45|-45\rangle$
4	0.0894${}^{\left(3.95\right)}$	0.0966${}^{\left(1.68\right)}$	0.1028${}^{\left(1.78\right)}$	0.1058${}^{\left(2.72\right)}$	0.1049${}^{\left(0.87\right)}$	0.1010${}^{\left(1.00\right)}$	0.0962${}^{\left(0.21\right)}$
6	0.0882${}^{\left(2.56\right)}$	0.0957${}^{\left(0.74\right)}$	0.1016${}^{\left(0.59\right)}$	0.1044${}^{\left(1.36\right)}$	0.1037${}^{\left(-0.29\right)}$	0.1005${}^{\left(0.50\right)}$	0.0960${}^{\left(0.00\right)}$
8	0.0868${}^{\left(0.93\right)}$	0.0948${}^{\left(-0.21\right)}$	0.1011${}^{\left(0.10\right)}$	0.1040${}^{\left(0.97\right)}$	0.1035${}^{\left(-0.48\right)}$	0.1004${}^{\left(0.40\right)}$	0.0959${}^{\left(-0.10\right)}$
10	0.0866${}^{\left(0.70\right)}$	0.0946${}^{\left(-0.42\right)}$	0.1009${}^{\left(-0.10\right)}$	0.1039${}^{\left(0.87\right)}$	0.1035${}^{\left(-0.48\right)}$	0.1004${}^{\left(0.40\right)}$	0.0959${}^{\left(-0.10\right)}$

**Table 4 materials-16-01395-t004:** Convergence of nondimensional energy for different ps−refinement strategies.

		Linear ps-Refinement	Uniform ps-Refinement	Graded ps-Refinement (m=1)
		$p_{s}=1$	$p_{s}=p$	$p_{s}=p$ − Floor$\left(\frac{s}{m}\right)$
p=1		52.4124	52.4570	52.3428
p=2		52.5307	52.5348	52.5254
p=3		52.5433	52.5439	52.5433
p=4		52.5445	52.5446	52.5445
p=5		52.5446	52.5446	52.5446
p=6		52.5446	52.5446	52.5446
Reference	52.5446			

**Table 5 materials-16-01395-t005:** Nomenclature for finite element models.

	Refinement Strategy	Mesh Resolution (*h*)	Polynomial Order (*p*)	Superposition Levels (*s*)
*h*−model	*h*	Increased	Fixed	-
*p*−model	*p*	Fixed	Increased	-
ps−L−model	Linear ps	Fixed	Increased	Increased, $p_{s}=1$
ps−U−model	Uniform ps	Fixed	Increased	Increased, $p_{s}=p$
ps−G−model	Graded ps	Fixed	Increased	Increased, $p_{s}$ set with Equation (Equation 44)

**Table 6 materials-16-01395-t006:** Nondimensional frequency $\hat{\omega }$ and buckling load ${\hat{N}}_{xx}$ for a SSSS anisotropic plate using a uniform ps−refinement strategy.

	Number of Unknowns	$\hat{\omega }$	${\hat{N}}_{\mathrm{xx}}$
p=1	1119	24.9319	52.5977
p=2	2714	22.0819	31.6995
p=3	4999	21.9341	30.2107
p=4	7974	21.9289	30.1487
p=5	11,639	21.9267	30.1416
p=6	15,994	21.9264	30.1407
p=7	21,039	21.9263	30.1402

**Table 7 materials-16-01395-t007:** Nondimensional frequencies $\hat{\omega }$ and buckling load ${\hat{N}}_{xx}$ for SSSS plates with different degrees of anisotropy.

	$E_{11}/E_{22}$	$\hat{\omega }$			${\hat{N}}_{\mathrm{xx}}$		
		ps-FEM	Ritz Method	%diff	ps-FEM	Ritz Method	%diff
θ=30	73.64	22.6796	22.6929	0.0584	38.7882	38.8328	0.1148
	40	17.9948	17.9913	0.0195	26.7974	26.7858	0.0432
	20	14.0520	14.0524	0.0032	17.7050	17.7064	0.0078
	10	11.1987	11.1987	0.0004	11.8977	11.8978	0.0007
θ=45	73.64	21.9263	21.9674	0.1873	30.1402	30.2538	0.3754
	40	17.6666	17.6557	0.0619	22.9941	22.9633	0.1339
	20	13.9480	13.9496	0.0115	16.4120	16.4163	0.0262
	10	11.1537	11.1538	0.0010	11.4625	11.4628	0.0024
θ=60	73.64	22.6796	22.6929	0.0584	24.1328	24.1571	0.1005
	40	17.9948	17.9913	0.0195	19.3018	19.2959	0.0307
	20	14.0520	14.0524	0.0032	15.0556	15.0563	0.0050
	10	11.1987	11.1987	0.0004	11.7175	11.7176	0.0010

**Table 8 materials-16-01395-t008:** Summary of the refinement parameters of the FE models used for solving the application example 3.

	Refinement	Number of Unknowns	*h*	*p*	*s*
*h*−model		91	2 × 2	1	-
	1	343	4 × 4	1	-
	2	1327	8 × 8	1	-
*p*−model		91	2 × 2	1	-
	1	343	2 × 2	3	-
	2	1021	2 × 2	6	-
ps−L−model		91	2 × 2	1	0
	1	252	2 × 2	2	5
	2	453	2 × 2	3	10

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
