# Peer review of "Application of the ps−Version of the Finite Element Method to the Analysis of Laminated Shells"

_materials, 2023, doi:10.3390/ma16041395_

Round 1

Reviewer 2 Report

The paper is well organized; however, there are still some concerns in the text.

In local responses, what is the accuracy of the advanced refinement?

What is the level of accuracy for 3D modeling?

Reviewer 3 Report

The manuscript may be accepted if the following errors are corrected:

1. Equation (40) is a nonlinear equation, does this work use Newton Raphson's method? Authors need to explain more how to solve?

2. Section 4.1, the authors need to explain why use grid with 5 elements? for the finite element method, need to investigate the convergence of the solution on the number of elements?

3. Formula (45): it is necessary to replace the symbol h with t to be consistent with the calculation theory part

4. The examples in section 4.2 are calculated with very thin plates and shells, a/t = 80-100, but the article uses the theory of first-order shear strain, the authors need further explanation.

5. To enrich the introduction section, the authors should refer to some of the following related publications:

-  https://doi.org/10.1140/epjp/s13360-022-02631-9

- https://doi.org/10.1007/s00419-021-02048-3

- DOI: 10.1007/978-981-10-7149-2_3
